# Modelling dynamics in protein crystal structures by ensemble refinement

**B Tom Burnley[1], Pavel V Afonine[2], Paul D Adams[2,3], Piet Gros[1]***

[1]Crystal and Structural Chemistry, Bijvoet Center for Biomolecular Research, Department of Chemistry, Faculty of Science, Utrecht University, Utrecht, The Netherlands; [2]Lawrence Berkeley National Laboratory, Berkeley, United States; [3]Department of Bioengineering, University of California Berkeley, Berkeley, United States

**Abstract** Single-structure models derived from X-ray data do not adequately account for the inherent, functionally important dynamics of protein molecules. We generated ensembles of structures by time-averaged refinement, where local molecular vibrations were sampled by molecular-dynamics (MD) simulation whilst global disorder was partitioned into an underlying overall translation–libration–screw (TLS) model. Modeling of 20 protein datasets at 1.1–3.1 Å resolution reduced cross-validated $R_{free}$ values by 0.3–4.9%, indicating that ensemble models fit the X-ray data better than single structures. The ensembles revealed that, while most proteins display a well-ordered core, some proteins exhibit a 'molten core' likely supporting functionally important dynamics in ligand binding, enzyme activity and protomer assembly. Order–disorder changes in HIV protease indicate a mechanism of entropy compensation for ordering the catalytic residues upon ligand binding by disordering specific core residues. Thus, ensemble refinement extracts dynamical details from the X-ray data that allow a more comprehensive understanding of structure–dynamics–function relationships.

*****For correspondence:
p.gros@uu.nl

**Competing interests:** The authors have declared that no competing interests exist

**Reviewing editor**: Axel T Brunger, Howard Hughes Medical Institute, Stanford University, United States

## Introduction

Since the dawn of structural biology there have been experimental observations of dynamic motion in proteins and other biomolecules (*Linderstrøm-Lang and Schellman, 1959*). Multiple biophysical methods have firmly established that such atomic 'wigglings and jigglings' (*Feynman et al., 1963*) play an inherent role in both protein structure and function; and, in conjunction with high-resolution structures insights into dynamics aid the understanding of biomolecular functions in catalysis, ligand or drug binding and macromolecular interactions. Presently X-ray diffraction and NMR spectroscopy are the primary source of data for high-resolution protein structures. Whereas microscopy methods may provide information regarding long-range conformational changes, NMR characterizes fluctuations at atomic detail. However, due to the challenging nature of such experiments the number of dynamics studies is relatively sparse in contrast with the wealth of structural information available in the Protein Data Bank (PDB) (*Berman et al., 2000*). The majority of entries in the PDB derived from X-ray diffraction data are presented as static, single, structures, although there is often extensive disorder resulting from protein dynamics and crystal-lattice distortions. Extracting atomic fluctuations from these diffraction data would dramatically increase the scope for dynamics studies of biomolecules and potentially reveal atomic details of structure–function–dynamic mechanisms that have previously been obscured.

The diffraction data of proteins are affected by multiple sources of disorder, notably arising from atomic vibrations, concerted motions of protein domains and inter-molecular lattice distortions. Structural models of proteins should account for both anisotropic and anharmonic distributions around the mean atomic positions to reproduce the observed Bragg intensities accurately (*Vitkup et al., 2002*; *Furnham et al., 2006*). However, explicit modelling of such distributions in macromolecules using current methods requires extensive parameterization inappropriate for the diffraction quality of

**eLife digest** It has been clear since the early days of structural biology in the late 1950s that proteins and other biomolecules are continually changing shape, and that these changes have an important influence on both the structure and function of the molecules. X-ray diffraction can provide detailed information about the structure of a protein, but only limited information about how its structure fluctuates over time. Detailed information about the dynamic behaviour of proteins is essential for a proper understanding of a variety of processes, including catalysis, ligand binding and protein–protein interactions, and could also prove useful in drug design.

Currently most of the X-ray crystal structures in the Protein Data Bank are 'snap-shots' with limited or no information about protein dynamics. However, X-ray diffraction patterns are affected by the dynamics of the protein, and also by distortions of the crystal lattice, so three-dimensional (3D) models of proteins ought to take these phenomena into account. Molecular-dynamics (MD) computer simulations transform 3D structures into 4D 'molecular movies' by predicting the movement of individual atoms.

Combining MD simulations with crystallographic data has the potential to produce more realistic ensemble models of proteins in which the atomic fluctuations are represented by multiple structures within the ensemble. Moreover, in addition to improved structural information, this process—which is called ensemble refinement—can provide dynamical information about the protein. Earlier attempts to do this ran into problems because the number of model parameters needed was greater than the number of observed data points. Burnley et al. now overcome this problem by modelling local molecular vibrations with MD simulations and, at the same time, using a course-grain model to describe global disorder of longer length scales.

Ensemble refinement of high-resolution X-ray diffraction datasets for 20 different proteins from the Protein Data Bank produced a better fit to the data than single structures for all 20 proteins. Ensemble refinement also revealed that 3 of the 20 proteins had a 'molten core', rather than the well-ordered residues core found in most proteins: this is likely to be important in various biological functions including ligand binding, filament formation and enzymatic function. Burnley et al. also showed that a HIV enzyme underwent an order–disorder transition that is likely to influence how this enzyme works, and that similar transitions might influence the interactions between the small-molecule drug Imatinib (also known as Gleevec) and the enzymes it targets. Ensemble refinement could be applied to the majority of crystallography data currently being collected, or collected in the past, so further insights into the properties and interactions of a variety of proteins and other biomolecules can be expected.

a typical protein crystal. Multi-conformer structures represent both anisotropic and anharmonic disorder, but despite numerous attempts at automating the inclusion of minor conformations (*DePristo et al., 2004*; *Levin et al., 2007*; *Terwilliger et al., 2007*; *Korostelev et al., 2009*; *van den Bedem et al., 2009*; *Lang et al., 2010*), 95% of all protein residues in the Protein Data Bank (PDB) (*Berman et al., 2000*) derived from diffraction data are modelled with a single conformation (*Lang et al., 2010*). As opposed to multiple discrete models, a MD simulation with time-averaged restraints (*Gros et al., 1990*) results in a population of structures in which the individual models are interrelated by a Boltzmann-weighted energy function. This method introduced by *Torda et al. (1989)* and implemented in macromolecular crystallography by *Gros et al. (1990)*, showed a reduction in $R$-value. However, cross-validation introduced subsequently (*Brünger, 1992*) revealed chronically over-fitted models (*Burling and Brunger, 1994*; *Clarage and Phillips, 1994*; *Schiffer et al., 1995*).

Here, we present an ensemble-refinement method that restricts the number of structures modelled and thereby prevents over-fitting of the data. We model large-scale motions, attributable to, for example, lattice distortions, by an underlying global disorder model. This approach allows MD simulations to sample local atomic fluctuations only, without the need for sampling large-scale global disorder. We show that the method yields reproducible ensembles with improved fit to the X-ray data, as validated by cross validation, $R_{free}$ (*Brünger, 1992*), and stereochemical analyses. Analyses of the ensembles show that detailed features are observed indicating atomic fluctuations that may be relevant for the biological function of the macromolecules.

# Results and discussion

## Ensemble refinement of 20 datasets from the PDB

We performed MD simulations, in which the model was restrained by a time-averaged X-ray (*Gros et al., 1990*), maximum-likelihood (*Pannu and Read, 1996*; *Adams et al., 1997*; *Murshudov et al., 1997*) target function (see 'Materials and methods'). The X-ray restraint optimized $\langle F_{calc}(hkl)\rangle$ against $F_{obs}(hkl)$, where $\langle F_{calc}(hkl)\rangle$ are computed as rolling averages from the structures in the MD trajectory, with the length of the averaging window determined by the relaxation time $\tau_x$. This approach contrasts with the traditional crystallographic refinement approach, where $F_{calc}(hkl)$ are computed from a single structure and optimized against $F_{obs}(hkl)$.

Prior to the simulations we approximated the large-scale disorder by an overall TLS model derived from the atomic *B*-factors of the refined single structure. Using one TLS group per protein molecule or domain, we iteratively fitted TLS parameters (*Schomaker and Trueblood, 1968*; *Winn et al., 2001*) to the atomic *B*-factors of the protein atoms excluding atoms with large deviations in *B*-factor from the

**Table 1.** Ensemble refinement statistics for 20 datasets. Datasets were taken from the PDB or PDB_REDO and were re-refined using ensemble refinement and phenix.refine. The relaxation time $\tau_x$ used, the resulting number of structures in the final ensemble and $R_{work}$ and $R_{free}$ values are given. The ensemble models yield improved $R_{free}$ values for all datasets, ranging in improvement from 0.3% to 4.9% with a mean improvement of 1.8%. The PDB accession numbers are as follows: 1KZK (*Reiling et al., 2002*), 3K0M (*Fraser et al., 2009*), 3K0N (*Fraser et al., 2009*), 2PC0 (*Heaslet et al., 2007*), 1UOY (*Olsen et al., 2004*), 3CA7 (*Klein et al., 2008*), 2R8Q (*Wang et al., 2007*), 3QL0 (*Bhabha et al., 2011*), 1X6P (*Dunlop et al., 2005*), 1F2F (*Kimber et al., 2000*), 3QL3 (*Bhabha et al., 2011*), 1YTT (*Burling et al., 1996*), 3GWH (*Rodríguez et al., 2009*), 1BV1 (*Gajhede et al., 1996*), 1IEP (*Nagar et al., 2002*), 2XFA (*Singh et al., 2011*), 3ODU (*Wu et al., 2010*), 1M52 (*Nagar et al., 2002*), 3CM8 (*He et al., 2008*) and 3RZE (*Shimamura et al., 2011*)

| PDB ID | Resolution (Å) | Ensemble refinement | | | | phenix.refine | | Ensemble—phenix.refine | |
|---|---|---|---|---|---|---|---|---|---|
| | | $\tau_x$ (ps) | No. of structures | $R_{work}$ | $R_{free}$ | $R_{work}$ | $R_{free}$ | $\Delta R_{work}$ | $\Delta R_{free}$ |
| 1KZK | 1.1 | 1.5 | 600 | 0.125 | 0.153 | 0.136 | 0.155 | −0.011 | −0.003 |
| 3K0M | 1.3 | 2.0 | 250 | 0.104 | 0.129 | 0.116 | 0.132 | −0.012 | −0.003 |
| 3K0N | 1.4 | 1.0 | 209 | 0.115 | 0.133 | 0.119 | 0.143 | −0.004 | −0.010 |
| 2PC0 | 1.4 | 0.8 | 250 | 0.145 | 0.188 | 0.161 | 0.193 | −0.016 | −0.005 |
| 1UOY | 1.5 | 1.0 | 167 | 0.104 | 0.137 | 0.155 | 0.185 | −0.051 | −0.049 |
| 3CA7 | 1.5 | 0.8 | 40 | 0.149 | 0.184 | 0.171 | 0.212 | −0.022 | −0.029 |
| 2R8Q | 1.5 | 1.0 | 200 | 0.132 | 0.162 | 0.158 | 0.178 | −0.026 | −0.016 |
| 3QL0 | 1.6 | 0.5 | 70 | 0.204 | 0.254 | 0.229 | 0.270 | −0.024 | −0.017 |
| 1X6P | 1.6 | 1.0 | 400 | 0.121 | 0.149 | 0.140 | 0.175 | −0.019 | −0.026 |
| 1F2F | 1.7 | 0.8 | 143 | 0.128 | 0.168 | 0.160 | 0.198 | −0.032 | −0.031 |
| 3QL3 | 1.8 | 0.5 | 80 | 0.160 | 0.208 | 0.170 | 0.221 | −0.010 | −0.013 |
| 1YTT | 1.8 | 0.3 | 84 | 0.139 | 0.174 | 0.166 | 0.189 | −0.027 | −0.014 |
| 3GWH | 2.0 | 1.0 | 39 | 0.160 | 0.200 | 0.187 | 0.220 | −0.027 | −0.021 |
| 1BV1 | 2.0 | 0.4 | 78 | 0.149 | 0.182 | 0.154 | 0.205 | −0.005 | −0.023 |
| 1IEP | 2.1 | 0.5 | 200 | 0.183 | 0.238 | 0.196 | 0.245 | −0.012 | −0.007 |
| 2XFA | 2.1 | 1.0 | 100 | 0.171 | 0.217 | 0.184 | 0.244 | −0.013 | −0.027 |
| 3ODU | 2.5 | 0.3 | 50 | 0.208 | 0.269 | 0.219 | 0.281 | −0.010 | −0.012 |
| 1M52 | 2.6 | 0.5 | 50 | 0.161 | 0.211 | 0.168 | 0.228 | −0.007 | −0.017 |
| 3CM8 | 2.9 | 0.5 | 67 | 0.194 | 0.235 | 0.205 | 0.248 | −0.011 | −0.013 |
| 3RZE | 3.1 | 0.1 | 72 | 0.210 | 0.280 | 0.210 | 0.291 | 0.000 | −0.011 |
| | | | | | | | **Max** | −0.051 | −0.049 |
| | | | | | | | **Min** | 0.000 | −0.003 |
| | | | | | | | **Mean** | −0.018 | −0.018 |

TLS-derived $B$-factor (the parameter $p_{TLS}$ described the percentage of atoms included in TLS-fitting; see 'Materials and methods'). The resulting TLS model was applied to all atoms throughout the simulation. Effectively, this TLS model of the protein core excludes the effects of hyper-flexible surface loops and, hence, describes the global disorder that may be attributed to inter-molecular lattice distortions and overall intra-molecular breathing or domain shifts.

Ensemble refinement was tested using 20 diffraction datasets from the PDB and started from either the PDB or PDB_REDO (*Joosten et al., 2010*) structures ('Materials and methods'). Upper resolution limits of the datasets ranged from 1.1 to 3.1 Å resolution and structures had 50 to 1,004 amino-acid residues in the asymmetric unit (*Table 1*). The simulations were run at an effective temperature of 300 K for the protein atoms, using a temperature bath ($T_{bath}$) slightly below 300 K to allow for heating due to the non-conservative nature of the time-averaged X-ray restraint modulated by its weight $w_{x-ray}$ ('Materials and methods'). Explicitly modelled solvent atoms were added and/or removed intermittently during the simulation dependent on the corresponding electron-density and difference maps ('Materials and methods'). Bulk solvent effects were accounted for by an averaged Flat Bulk-Solvent Model (*Jiang and Brünger, 1994*; *Afonine et al., 2005*) ('Materials and methods'). The parameters $p_{TLS}$, $\tau_x$ and the $T_{bath}$ and $w_{x-ray}$ pair were optimized in a grid search resulting in a shallow optimum scored by $R_{free}$ (*Figure 1A*). After a period of equilibration, the trajectory of structures was acquired over an extensive period of time (40 times $\tau_x$).

*Figure 1B* shows the $R$-values as they developed over the simulation time for a structure with PDB code 1UOY (*Olsen et al., 2004*), for which the largest improvement in $R_{free}$ was observed among the datasets tested (possibly due to the high degree of anisotropic and anharmonic side-chain motion for this case). The $R$-values started at a high value and remained high (~35%) for the individual structures, which is in agreement with the observation that the derived global TLS $B$-factor model is not optimal for fitting a single structure to the data. Averaging the structure factors over the relaxation time $\tau_x$ of 1 ps (corresponding to the rolling average structure factors used in the X-ray restraint) dropped the $R_{work}$ and $R_{free}$ to ~11% and ~15% respectively. The $R_{work}$ and $R_{free}$ of the collected ensemble of structures (corresponding to unweighted averaged structure factors) monotonically decreased to 10.3 and 13.7% respectively. over the acquisition period of 40 ps. The improvement in $R$-values from the ensemble model with respect to the single-structure model spanned the entire resolution range of the data (*Figure 1C*). Acquisition over 40 times $\tau_x$ yielded a highly redundant set of structures. We reduced the number of structures by calculating the minimum number of structures, that is 167 in the case of 1UOY, required to reproduce the $R$-value of the trajectory (*Figure 1D* and 'Materials and methods').

Analysis of all 20 datasets showed that ensemble refinement improved the $R_{free}$ by between 0.3 and 4.9 percentage points compared to single structures re-refined using the same program package, that is Phenix (*Afonine et al., 2012*), with a mean improvement of 1.8% in $R_{free}$ values (*Table 1*, *Figure 2A*). The effect of the starting structure on ensemble refinement was assessed by using alternative refinement programs, phenix.refine, Refmac (*Vagin et al., 2004*), and Buster (*Bricogne et al., 2009*), to generate varying input models. No significant differences were observed due to the different starting models (*Tables 3 and 4*). The improvement in $R_{free}$, number of structures in the final ensemble and the averaging time $\tau_x$ tended to increase with resolution (*Figure 2A–C*). The optimum values for the parameters $p_{TLS}$ and $T_{bath}$ are not correlated with resolution (*Figure 2D,E*). Concomitant with the reduction in $R$-values, the ensemble models reduced electron-density differences, decreasing rms fluctuations in difference maps by 0 to 41% with an average of 12% improvement (*Table 2*). The difference electron-density maps for the single-structure and ensemble models indicated improvements throughout the asymmetric unit cell, as exemplified in *Figure 1E*.

## Validation of ensemble refinement

We used the high-quality experimental phases available to high resolution for 1YTT of mannose-binding protein (*Burling et al., 1996*) to validate the ensemble-refinement method. The overall correlation coefficient between the electron-density map from the ensemble model (obtained without experimental phases) and the experimentally phased electron-density map was 0.903, compared with 0.873 and 0.895 for the published and re-refined single structures. These seemingly small improvements in overall quality indicators allow for significant local improvements. Real-space correlation coefficients (*Brändén and Jones, 1990*) highlighted marked local improvements for flexible residues in particular (*Figure 3A*) with 11 residues improving by more than 0.1 in correlation coefficient. This observation was consistent with local improvements in electron-density differences in regions of flexible or disordered

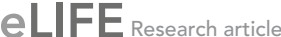

**Figure 1**. Example of ensemble refinement for dataset 1UOY. (**A**) Optimisation of empirical ensemble refinement parameters ($\tau_x$, $p_{TLS}$ and $T_{bath}$). Simulations are performed independently and in parallel. The plot shows effect of $\tau_x$, $p_{TLS}$ on $R_{free}$ (each grid point corresponds to the lowest $R_{free}$ among all $T_{bath}$ values). Optimum parameters are selected by $R_{free}$. (**B**) R-values obtained during ensemble-refinement simulation, solid lines $R_{work}$ and dashed lines $R_{free}$; high values are observed for instantaneous models (yellow) contrasting with the rolling average used in the target function (red) and the final ensemble (blue). (**C**) R-values are reduced throughout the resolution range for ensemble model (blue) compared with phenix.refine re-refined single structure (black); solid lines $R_{work}$ and dashed line $R_{free}$. (**D**) Number of structures in the ensemble, reduced by equidistant selection, *versus* $R_{work}$ (solid line) and $R_{free}$ (dashed line). Final number of structures is selected as the minimum number required reproducing the $R_{free}$ + 0.1%; in this case resulting in an ensemble containing 167 structures. (**E**) Density difference maps for the ensemble structure ($mF_{obs} - DF_{model}$)exp[$i\varphi_{model}$], left-hand side, and the single structure right-hand side, contoured at 0.34 e/Å³ (equivalent to 3.0 σ for the ensemble model), positive and negative densities are coloured green and red respectively. All molecular graphics figures are drawn using PyMol (The PyMOL Molecular Graphics System, Schrödinger, LLC).

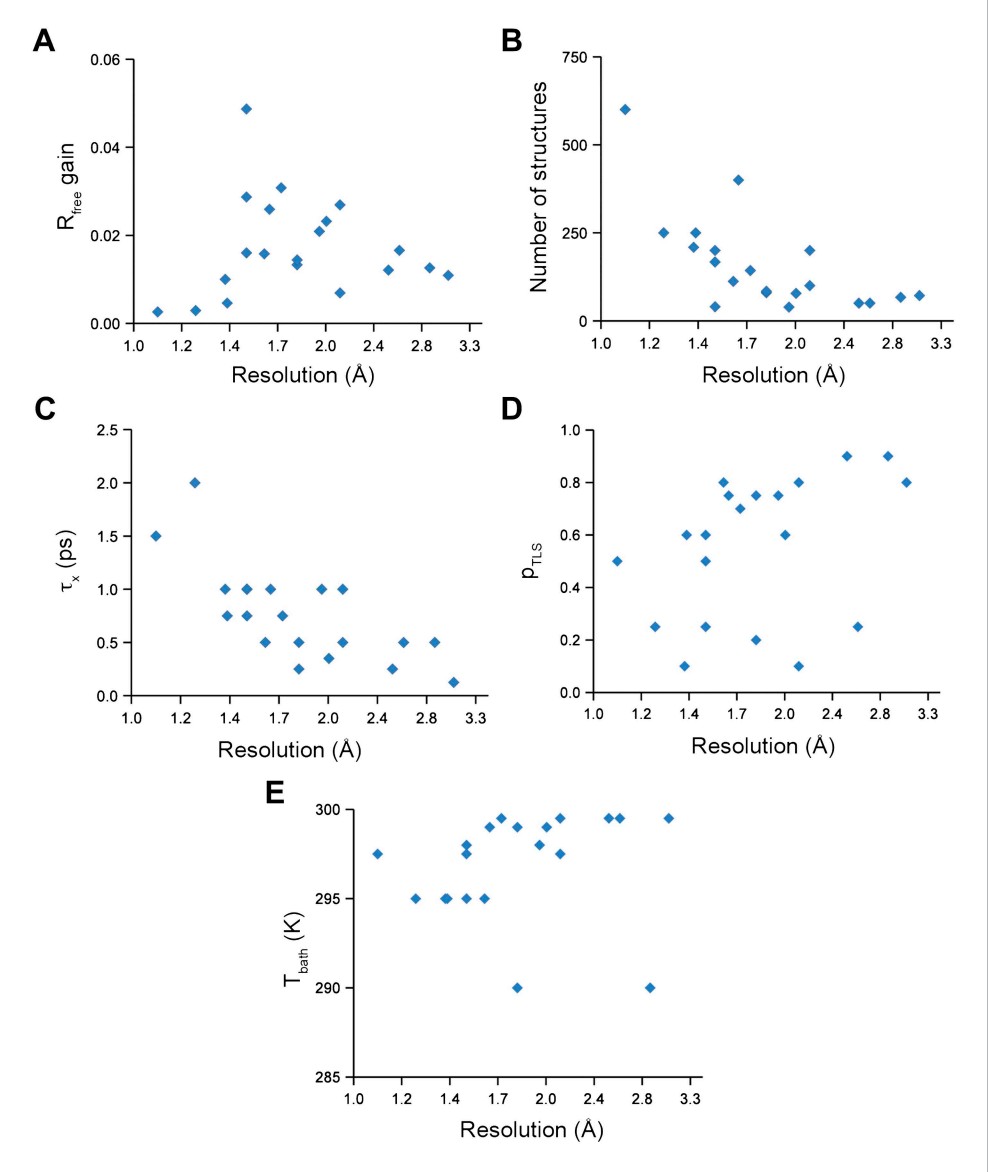

**Figure 2**. Ensemble refinement parameters and results as function of resolution of the datasets. (**A**) Gain in $R_{free}$ of ensemble refinement compared with re-refinement using phenix.refine, (**B**) number of structures in the final ensemble model, (**C**) optimum relaxation time, $\tau_x$, (**D**) optimum $p_{TLS}$ and (**E**) optimum $T_{bath}$ plotted as function of resolution of the dataset.

side chains (**Figure 3B**). Moreover, the ensemble model contained structural details previously identified in a multiple-model approach by **Burling et al. (1996)**, as shown for the anisotropic distribution for the side chain of Phe121 (**Figure 3C**) and diffuse water shells around hydrophobic residues (**Figure 3D**). **Figure 3E** shows that the most occupied water sites in the ensemble correlated with low atomic *B*-factors for waters in the single-structure model.

Next, we analysed the stereochemistry of the computed ensemble models. The robustness of the observed atomic distributions was tested by repeating ensemble refinements 10 times using different random starting velocities. **Figure 4A** shows that the observed distributions are highly reproducible. With data extending to 1.5-Å resolution correlations above 0.99 were observed between residue distributions from separate runs. At lower resolutions, the majority of residues showed correlations above 0.95, with occasionally correlations dropping below 0.8 in very flexible regions (see **Figure 4B,C**). Clearly, the ensembles contain multiple values for each geometrical term that form a distribution,

**Table 2.** Rms $(mF_{obs} - DF_{model})\exp[i\varphi_{model}]$ difference densities obtained from ensemble refinement and re-refinement in phenix.refine

| PDB ID | Resolution (Å) | $\sigma_{mFo-DFc}$ (e/Å³) | |
| --- | --- | --- | --- |
| | | Ensemble | phenix.refine |
| 1KZK | 1.1 | 0.138 | 0.161 |
| 3K0M | 1.3 | 0.016 | 0.018 |
| 3K0N | 1.4 | 0.007 | 0.008 |
| 2PCO | 1.4 | 0.099 | 0.099 |
| 1UOY | 1.5 | 0.115 | 0.162 |
| 3CA7 | 1.5 | 0.132 | 0.148 |
| 2R8Q | 1.5 | 0.104 | 0.118 |
| 3QL0 | 1.6 | 0.124 | 0.138 |
| 1X6P | 1.6 | 0.098 | 0.105 |
| 1F2F | 1.7 | 0.104 | 0.126 |
| 3QL3 | 1.8 | 0.131 | 0.139 |
| 1YTT | 1.8 | 0.170 | 0.215 |
| 3GWH | 2.0 | 0.125 | 0.138 |
| 1BV1 | 2.0 | 0.109 | 0.119 |
| 1IEP | 2.1 | 0.084 | 0.091 |
| 2XFA | 2.1 | 0.069 | 0.074 |
| 3ODU | 2.5 | 0.105 | 0.113 |
| 1M52 | 2.6 | 0.088 | 0.093 |
| 3CM8 | 2.9 | 0.036 | 0.036 |
| 3RZE | 3.1 | 0.070 | 0.070 |

instead of a single stereochemical value obtained from a single structure; *Figure 5A–D* presents examples of side-chain distributions (by $\chi_1$ and $\chi_2$ angles) observed in the ensembles along with standard deviations computed from the 10 repeats. Averaged over all 20 cases, the rms deviations from idealized bond lengths, bond angles and dihedral angles for the re-refined single-structures were 0.010 Å, 1.26° and 15.2° respectively. (*Figure 6—source data 1*). These deviations decreased for the ensemble models by 0.002 Å, 0.26° and 6.6° respectively, when considering the centroids of the observed stereochemical distributions. Taking all fluctuations around the centroids (i.e. complete distributions) into account, these values increased by 0.002 Å, 0.33° and 4.0° respectively compared to the statistics from single structures. This indicates high stereochemical quality for the ensemble model, but that the ensemble of structures contained fluctuations exhibiting larger deviations from ideality. *Figure 6A,B* shows that high-energy conformations, as indicated by for example non-favourable Ramachandran φ,ψ-angle combinations, occurred transiently and were concentrated in regions of structural flexibility. Counting the most frequent Ramachandran classification for each φ,ψ-angle showed that the ensembles have a similar percentage for 'allowed' and an increased number of 'outliers' compared to the single structures (*Figure 6C* and *Figure 6—source data 2*). These analyses illustrate that in ensemble refinement conformations were sampled, rather than optimized to a single configuration as in single-structure refinement. Similar to Brünger (*Brünger, 1992*), we observe that lower $R_{free}$ values correlate with better quality of the Ramachandran statistics (*Figure 6D*).

The presence of non-crystallographic symmetry (NCS) allowed for crystallographically-independent observations of atomic fluctuations in multiple copies of a protein molecule (*Figure 7A–C*). In some cases, the applied global TLS models differed significantly between NCS-related copies (*Figure 7B*). Nevertheless, we observed atomic fluctuations similar both in magnitude and location for related copies in areas not affected by crystal packing (*Figure 7B,C*; additional cases of NCS are presented in *Figure 7—figure supplement 1–4*). Apparently, variations in overall disorder arising from packing differences of NCS copies (as indicated by different *B*-factor distributions) were well accounted for by the

**Table 3.** Effect of input structure on ensemble refinement. For three datasets ensemble refinement was performed using a starting structure from three different refinement programs. For each structure three random number seed repeats of ensemble refinement were performed and the *R*-factors are shown to be highly similar

| | | Re-refinement | | Ensemble refinement | | | | | | | | | |
| | | | | Repeat 1 | | Repeat 2 | | Repeat 3 | | Mean | |
| PDB | Program | $R_{work}$ | $R_{free}$ | $R_{work}$ | $R_{free}$ | $R_{work}$ | $R_{free}$ | $R_{work}$ | $R_{free}$ | $R_{work}$ | $R_{free}$ |
|---|---|---|---|---|---|---|---|---|---|---|---|
| 1UOY | Buster | 0.167 | 0.196 | 0.108 | 0.144 | 0.112 | 0.145 | 0.110 | 0.146 | 0.110 | 0.145 |
| | Refmac | 0.147 | 0.170 | 0.104 | 0.137 | 0.103 | 0.140 | 0.105 | 0.144 | 0.104 | 0.140 |
| | Phenix | 0.155 | 0.185 | 0.109 | 0.142 | 0.109 | 0.147 | 0.111 | 0.149 | 0.110 | 0.146 |
| 3CA7 | Buster | 0.177 | 0.208 | 0.137 | 0.186 | 0.137 | 0.192 | 0.141 | 0.197 | 0.138 | 0.192 |
| | Refmac | 0.170 | 0.205 | 0.139 | 0.187 | 0.135 | 0.189 | 0.138 | 0.193 | 0.137 | 0.189 |
| | Phenix | 0.171 | 0.212 | 0.138 | 0.180 | 0.142 | 0.189 | 0.148 | 0.193 | 0.142 | 0.187 |
| 1BV1 | Buster | 0.161 | 0.204 | 0.137 | 0.184 | 0.138 | 0.185 | 0.137 | 0.186 | 0.138 | 0.185 |
| | Refmac | 0.178 | 0.231 | 0.140 | 0.182 | 0.143 | 0.184 | 0.143 | 0.189 | 0.142 | 0.185 |
| | Phenix | 0.154 | 0.205 | 0.139 | 0.188 | 0.138 | 0.189 | 0.140 | 0.189 | 0.139 | 0.189 |

applied global TLS models. Similarly, a global increase in disorder present in a dataset collected at ambient temperature vs an isomorphous dataset collected under cryo-conditions was fully accounted for by an increase in global TLS (*Figure 8A*). These data indicate that the derived atomic fluctuations are molecular traits and that the global TLS model accounts for overall disorder, which includes for example lattice or packing effects.

## Functional dynamics revealed by ensemble refinement

Inspection of the obtained ensembles showed that most proteins, as expected, are characterized by well-ordered residues in the protein core and flexible residue side chains and loops on the outside (an example is given in *Figure 9A*). However, three cases exhibited marked flexibility of residue side chains on the inside of the molecule. 1BV1 (*Gajhede et al., 1996*), major birch pollen allergen, has a large forked solvent channel with multiple disordered side chains and a diffuse water network (*Figure 9B*). The cavity is consistent with its putative role as a general plant steroid carrier (*Marković-Housley et al., 2003*). Presumably, the flexible internal residues play a role in binding the diverse ligands. More surprising are the disordered cores in 1X6P of PAK pilin (*Dunlop et al., 2005*) (*Figure 9C*) and 3K0N of the enzyme proline isomerase (*Fraser et al., 2009*) (*Figure 9D*); in both these cases, the datasets were recorded at ambient temperatures. Multiple (16) aliphatic and aromatic side chains are highly flexible, forming a molten core in the pilin molecule. These flexible residues, which are extremely well conserved in the type IV pilin family, create the central interface between the characteristic long α-helix and β-sheet of this protein fold (*Hazes et al., 2000*). We hypothesize that this monomeric pilin structure represents an intermediate molten state, which becomes stabilized upon protomer filament formation. The third case with flexible residues on the interior is 3K0N of proline isomerase (*Fraser et al., 2009*). As with the pilin structure, several (11) aromatic and aliphatic residues showed large side chain fluctuations, yielding a molten core of the protein structure. However, the same protein under cryogenic conditions (3K0M) (*Fraser et al., 2009*) showed mostly well-ordered side chains in the ensemble (*Figure 9D*, right-hand side), indicating that at cryogenic conditions the molten core has been annealed to its ground state configuration. As discussed in more detail in the next paragraph, the observed flexibility of the core residues at ambient temperature is likely of functional relevance for the enzyme. Thus, the computed ensemble models highlighted a hitherto unnoticed phenomenon of molten cores in folded proteins, which are likely relevant for the biological function of these molecules.

NMR spectroscopy has previously revealed specific dynamics for active-site residues of proline isomerase (*Eisenmesser et al., 2005*). The solvent-exposed residues Arg55 and Met61 in the active site showed disorder in 3K0M (*Fraser et al., 2009*), where data were collected at 100 K. For 3K0N collected at 288 K, a number of additional residues with multiple conformations were observed (*Figure 8A,B*).

**Table 4.** $F_{model}$ cross-correlation scores for ensembles generated with different input models. Three different refinement programs generated alternative starting structures, see **Table 3**. The best ensemble was selected as judged by $R_{free}$. $F_{model}$ cross correlation scores are >0.99 for all pairs of ensemble structures for all three datasets

| | Ensemble pair | | |
|---|---|---|---|
| PDB | Re-refined input | Re-refined input | CC |
| 1UOY | Refmac | Buster | 0.997 |
| | Refmac | Phenix | 0.997 |
| | Buster | Phenix | 0.999 |
| 3CA7 | Refmac | Buster | 0.993 |
| | Refmac | Phenix | 0.992 |
| | Buster | Phenix | 0.996 |
| 1BV1 | Refmac | Buster | 0.992 |
| | Refmac | Phenix | 0.990 |
| | Buster | Phenix | 0.992 |

These included Ser99, Phe113, which are part of the substrate-binding pocket together with Arg55 and Met61 (**Figure 10A**), and Leu98, which neighbours the flexible residue Ser99 but points into the hydrophobic core (**Fraser et al., 2009**). Ensemble refinement of the 288 K data revealed a large number of residue side chains in the core to be flexible. This flexibility in the core appears to be linked to the dynamics of active-site residues through the intervening β-sheet. In particular, the main-chain H-bonding network (C=O·HN) of neighbouring β-strands within the sheet was flexible, as indicated by anisotropy in the C=O bonds of residues 55-62-113-98 (with the largest anisotropy observed for 55 and 62; see **Figure 10B**). Analysis of the side-chain conformations for the active-site residues Arg55, Met61, Ser99 and Phe113 showed a 10% minor conformation for the four active-site residues (**Figure 10C**), which is in good agreement with NMR relaxation data (see Figure 2 in **Eisenmesser et al., 2005**). Mutation of Ser99 to Thr (>14 Å away from the catalytic Arg55) affects the side-chain distributions and lowers the activity ~300-fold, similar to an Arg55Lys mutation of the catalytic residue (**Eisenmesser et al., 2005**; **Fraser et al., 2009**). Thus, the ensemble refinement results support the notion put forward by Eisenmesser et al. and Fraser et al. that side chain dynamics play a critical role in the enzymatic function of proline isomerase and, moreover, expand upon this theme to reveal mechanistic insights arising from the underlying detailed dynamics.

Ligand binding to HIV protease is known to have marked effects on the enzyme structure (**Heaslet et al., 2007**). We compared HIV protease in its apo form, 2PC0 (**Heaslet et al., 2007**), and bound to ligand JE-2147, 1KZK (**Reiling et al., 2002**). As for proline isomerase, HIV protease exhibited flexible, moldable, substrate-binding pockets in the apo state. Enthalpic and entropic binding of the ligand with high affinity ($K_D$ = 41 pM) (**Velazquez-Campoy et al., 2001**) reduced the flexibility in the substrate binding pockets by protein–ligand H-bond interactions and van der Waals stacking (**Figure 11A,B**). Upon ligand binding, Asp90 became ordered through H-bonding with the ligand in P2, whereas its dimeric partner lacked a H-bonding partner in P2′ and remained flexible as in the unbound state. The canonical aspartic protease catalytic residues, Asp25 of both monomers, became ordered upon ligand binding. Concomitantly, we observed significant changes in dynamics of specific core residues (**Figure 11C**). Some residues, most notably Thr26, 'froze' (Thr26 is part of the conserved Asp25-Thr26-Gly-27 sequence). In contrast, the side chains of other residues, most notably Cys95 and Leu97 opposite of Thr26, became highly disordered in the bound state, whereas they were relatively ordered in the unbound state. This observation supports NMR data that showed that conformational variability increases upon inhibitor binding for Leu97 amongst others (**Torchia and Ishima, 2003**). These data suggest that the entropy lost by the catalytic aspartates upon ligand binding is compensated with an increase in disorder of specific core residue side chains. This type of dynamic modulation was also observed for $Ca^{2+}$ binding to calmodulin, where this effect was dubbed entropy compensation (**Lee et al., 2000**). Similar to the molten core dynamics for proline isomerase, the structure ensembles

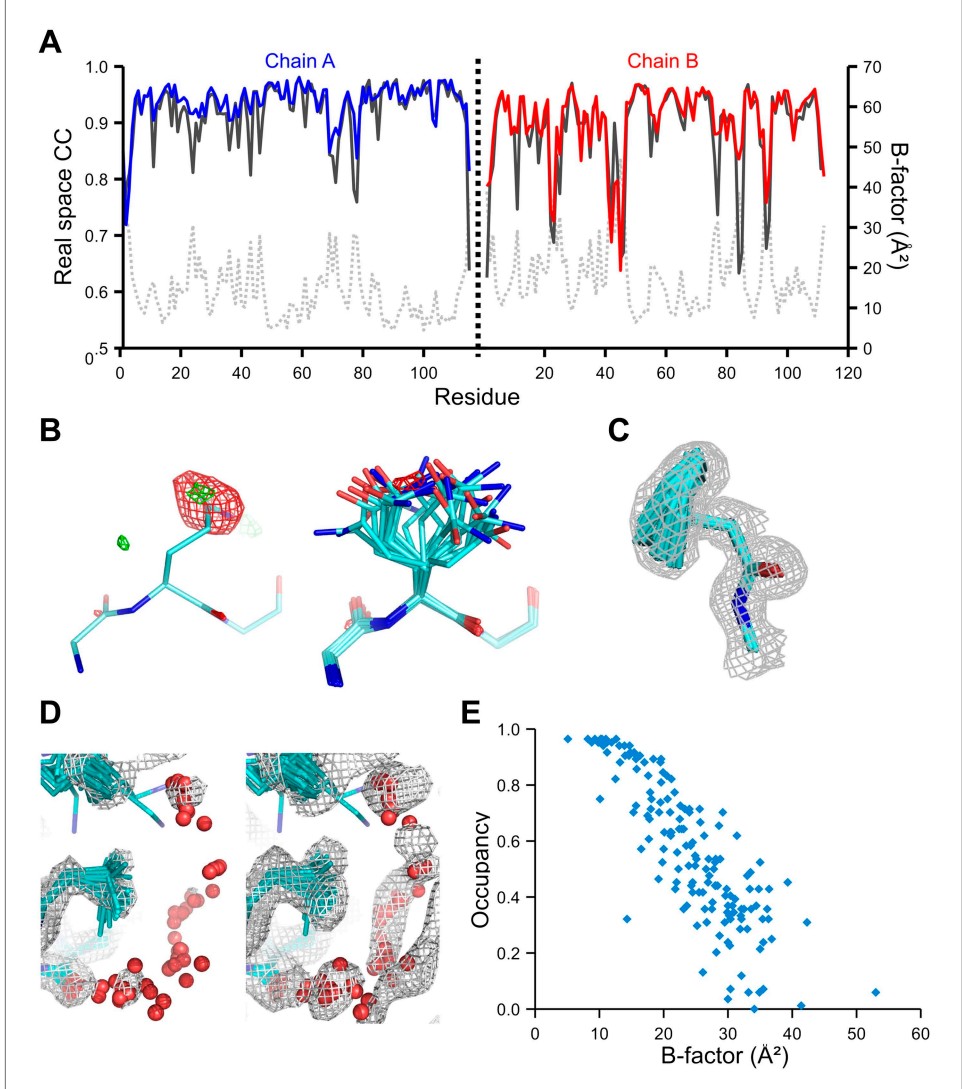

**Figure 3**. Validation of ensemble refinement using dataset 1YTT with exceptionally high quality experimental phases. (**A**) Real space cross-correlation of experimentally phased electron density map ($|F_{obs}|\exp[i\varphi_{obs}]$) and model map ($|F_{model}|\exp[i\varphi_{model}]$) for the single-structure (black) and ensemble model (chain A and B, blue and red respectively) shows improvements particularly for disordered areas (atomic *B*-factors from the re-refined single structure are shown in grey dashed lines). (**B**) Example of improved vector-difference map ($|F_{obs}|\exp[i\varphi_{obs}] - |F_{model}|\exp[i\varphi_{model}]$), contoured at 0.71 e/Å³ equivalent to 2.5 σ for the single structure for Gln167, chain A, for single (left-hand side) and ensemble structure (right-hand side). (**C**) Conformer distribution of Phe121 (chain A) with the experimental phased map ($|F_{obs}|\exp[i\varphi_{obs}]$) contoured at 1.4 σ is highly similar to the multi-conformer shown in Figure 1c in ***Burling et al. (1996)***. (**D**) Partially disordered solvent shell (red) around residue Leu203 (chain A) as anticipated in ***Burling et al. (1996)***. Ensemble structure with experimental phased experimental map ($|F_{obs}|\exp[i\varphi_{obs}]$) contoured at 1.4 σ (left side) and 0.7 σ (right side), as shown in Figure 2b in ***Burling et al. (1996)***. (**E**) Scatter plot showing the anti-correlation between the *B*-factor of explicit solvent molecules in the re-refined single-structure and the relative occupancy of water molecules at that same position (within 0.5-Å distance) in the ensemble model. Due to the difficulty in differentiating between disorder (*B*-factor) and occupancy for explicitly modelled water atoms in single structures a high *B*-factor is likely to correspond to a partially occupied site.

generated by the ensemble refinement method revealed specific core dynamics for HIV protease, in particular a conformational exchange that is likely functionally relevant.

The development of new small molecule therapeutics is often supported by the use of macromolecular structure, typically X-ray crystallography of complexes between target proteins and drug

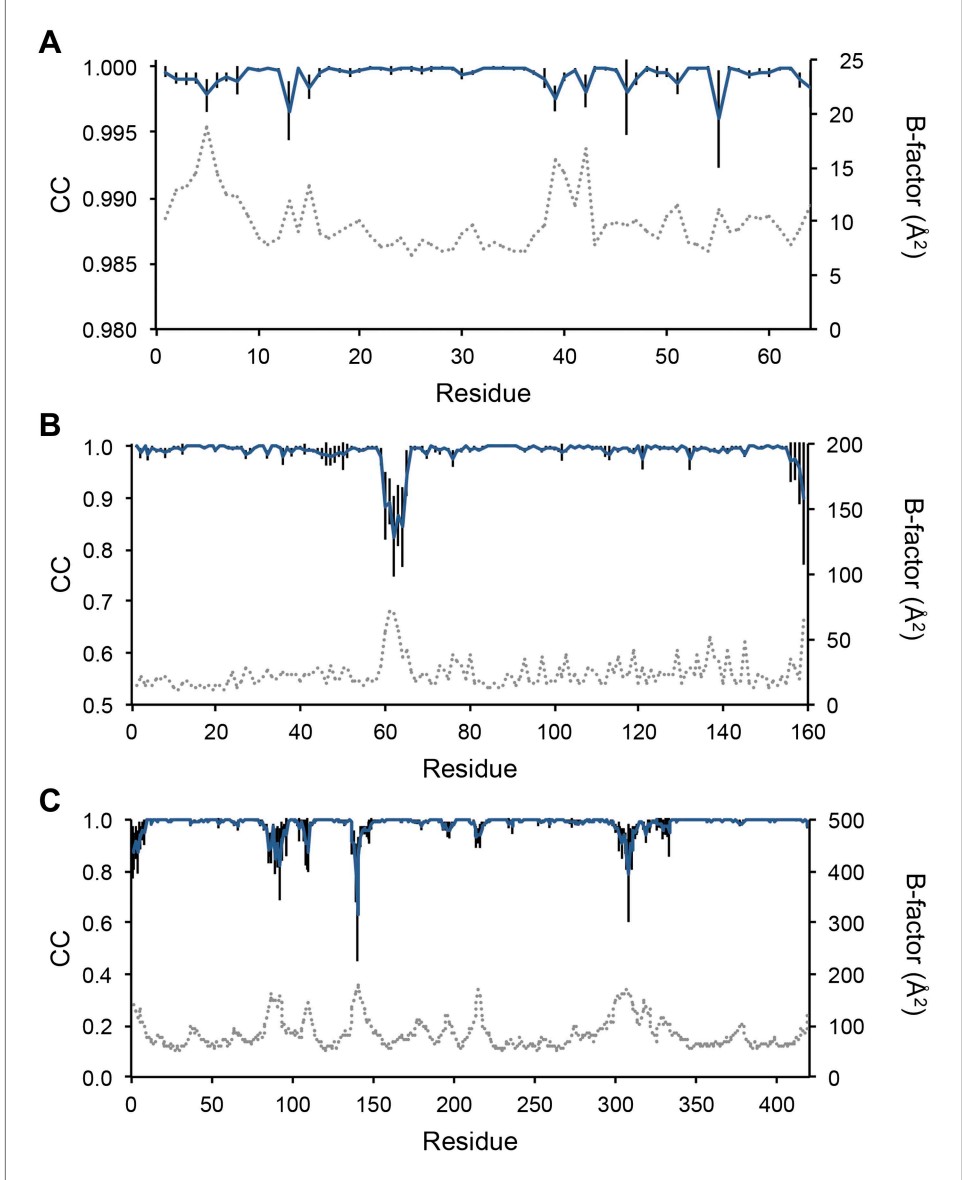

**Figure 4**. Sampling reproducibility of ensemble refinement. (**A**) Cross-correlations (CC) calculated for all pairs from 10 random-number seed repeat ensemble refinements of the 1UOY dataset extending to 1.5-Å resolution. (**B**) Cross correlations computed for 1BV1 (2.0-Å resolution); and, (**C**) for 3CM8 (2.9-Å resolution). Mean CC shown in solid blue (black error bars indicate ±1 *σ*). Cross correlations were computed from real-space *F*$_{model}$ electron-density map correlations (***Brändén and Jones, 1990***). *B*-factors from the single structures refined using phenix.refine are shown in dotted grey lines.

candidates. These complexes are typically interpreted as static structures, and the impact of dynamics, if considered at all, is probed using computational methods. Our new ensemble refinement approach makes it possible to study the role of dynamics in drug–target complexes in the context of the experimental data. Therefore, we analysed two structures of Abl kinase in complex with Imatinib (also known as Gleevec), that is 1IEP, and PD173955, that is 1M52 (***Nagar et al., 2002***). These compounds bind the Abl kinase with high affinity, 37 nM (***Schindler et al., 2000***) and 100 nM respectively (***Nagar et al., 2002***). The ensembles provide insights into the flexibility of the protein residues and the ligand moieties in the complex. ***Figure 12A*** shows the variation in H-bonding observed in Abl kinase–Imatinib. Variable H-bonding interactions were observed for the hydrophilic N-methylpiperazine moiety with the backbone carbonyl atoms of Ile360 and His361. In contrast, the ensemble displayed a

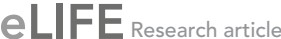

**Figure 5**. Reproducibility of side-chain rotamer distributions. Mean $\chi_1$ and $\chi_2$ distributions of four side-chains from the 10 repeats, with error bars ±1 $\sigma$, are shown for 1UOY. The four residues presented are those with the two highest CC values (see **Figure 4A**), (**A**) Gln11 (0.9999) and (**B**) Arg32 (0.9999), and the two lowest CC values, (**C**) Lys39 (0.9976) and (**D**) Arg13 (0.9966).

well-ordered H-bond between the anilino-NH and Thr315 'gatekeeper' side-chain. Moreover, the ordered water network between Glu286, Lys271 and the pyrimidine moiety of Imatinib (**Nagar et al., 2002**) was reproduced in the ensemble model (**Figure 12B**). We observed that the Abl kinase adopts two different states in these crystal structures. In the Imatinib complex the activation loop, residues 381–402, is highly disordered (**Figure 12C**), which was confirmed by comparison to previously published NMR data (**Vajpai et al., 2008**). In general, the ensemble models indicate details of tight and highly ordered drug–target interactions on one side vs disordered interactions elsewhere, which are indicative of less tight interactions, that may suggest which sites to modify in a drug-optimization cycle.

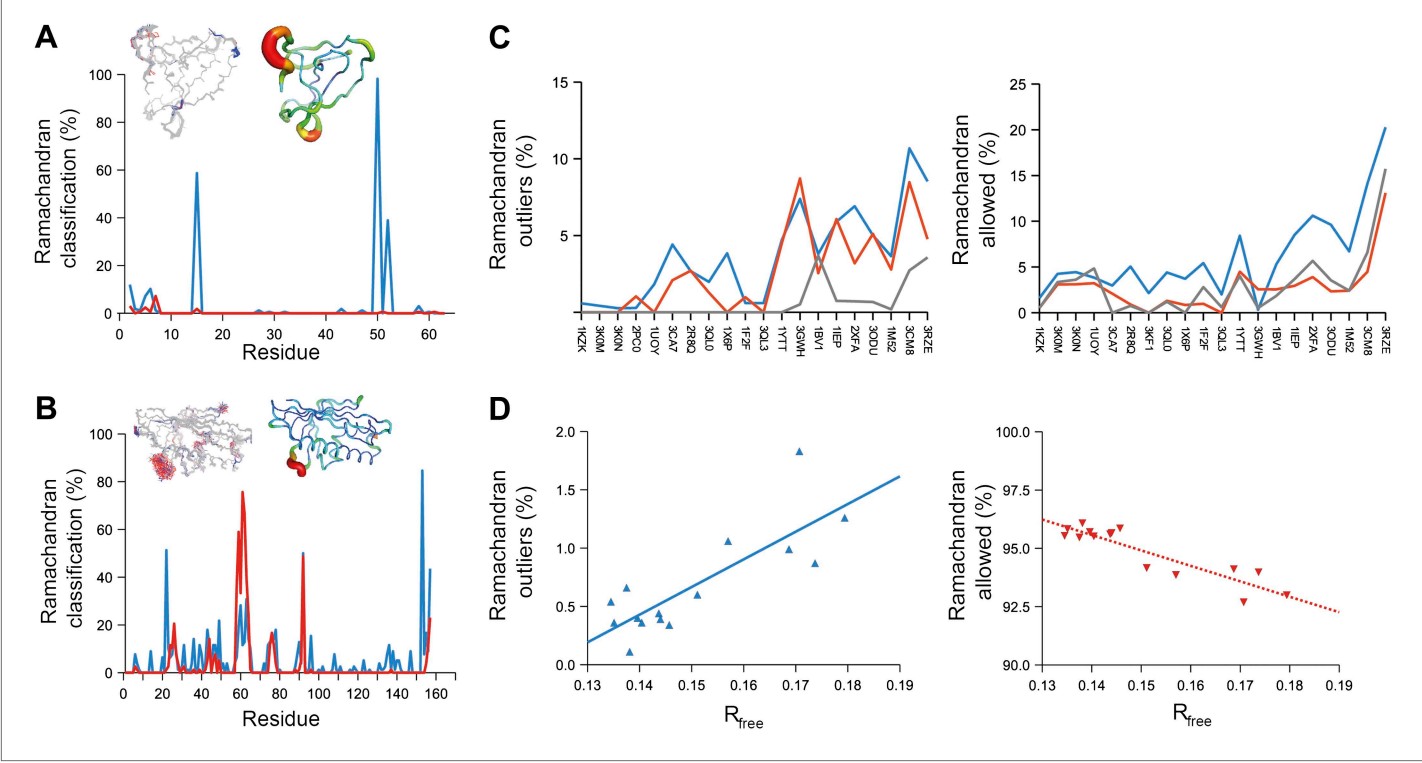

**Figure 6**. Ramachandran analysis. Distribution of Ramachandran torsion angles classified as outliers (red) and allowed (blue) for ensemble models, 1UOY (**A**) and 1BV1 (**B**). Plot shows percentage of classification per residue (i.e. relative number of times a φ,ψ-torsion angle combination is scored as outlier or allowed as defined by phenix.ramalyze). Structure inserts show (left-hand side) the location of the non-favourable torsion angles, outliers (red) and allowed (blue), and (right-hand side) a *B*-factor putty representation for the single structure refined with phenix.refine. (**C**) Overall Ramachandran statistics for ensemble and re-refined models. The Ramachandran statistics for the ensemble models are calculated in two ways: blue shows the percentage of outliers (left side) or allowed (right side) from all structures in the ensemble (cf. 'whole distribution' in **Figure 6—source data 1**), whereas red shows these percentages based on the most frequent occurring classification of each φ,ψ combination (cf. 'centroid distribution'). The grey lines show the percentage of allowed (left side) and outliers (right side) for the re-refined single structures. Ramachandran statistics per re-refined single structure and ensemble are given in **Figure 6—source data 2**. (**D**) Correlation of Ramachandran statistics with $R_{free}$ values obtained from ensemble refinement. Three ensemble refinements were performed for the dataset 1UOY using different random-number seeds at $T_{bath}$ values of 220, 260, 280, 290 and 295 K. Shown are the number of Ramachandran outliers (left side) and allowed (right side) in the ensemble as function of the $R_{free}$ value.

The following source data are available for figure 6.

**Source data 1.** Geometries of single-structure models and ensemble models.

**Source data 2.** Ramachandran statistics for re-refined and ensemble models.

## Conclusions

We have shown that far more structural information can be reliably extracted from protein diffraction data than is achieved to date by traditional single-structure modelling methods. Our ensemble refinement method samples distributions that reflect structural details of protein dynamics. The resulting ensemble models provide a more comprehensive description of the molecules and allow interpretation of the molecular function in terms of both the three-dimensional arrangements of the protein residues and their flexibilities. Moreover, ensemble models minimize the risk of structural over-interpretation associated with the seemingly rigid single-structure models. We found comparative analyses of protein molecules in different states to be very useful for identifying detailed changes in structural dynamics that may be mechanistically relevant for the molecular function.

Partitioning large-scale disorder into a global model separates intermolecular variations of protein packing in the crystal from the detailed intra-molecular atomic fluctuations. Effectively, the

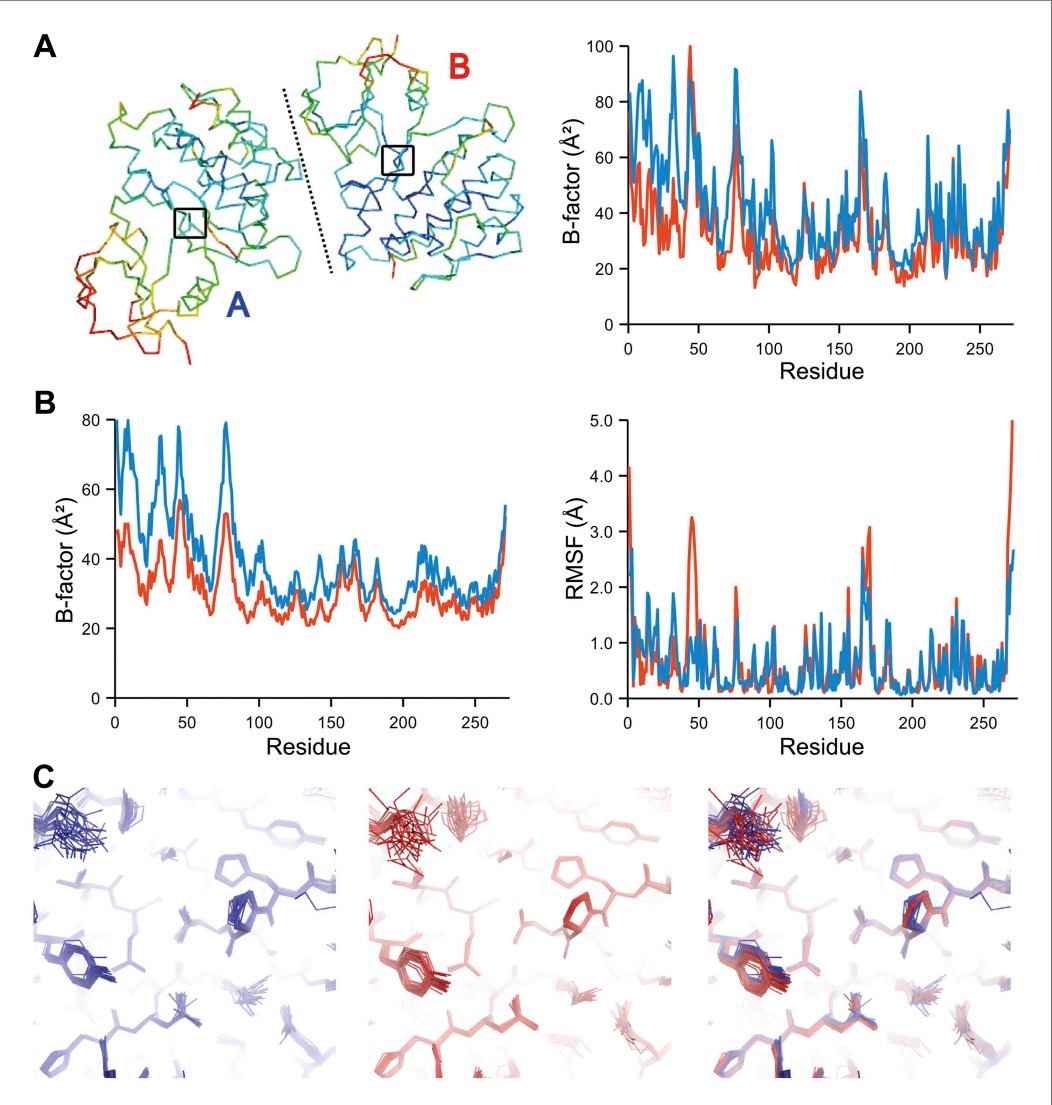

**Figure 7**. Comparison of atomic fluctuations for non-crystallographic symmetry related protein copies for dataset 1M52. (**A**) Cα trace of the re-refined single structure coloured by *B*-factor (from blue to red with increasing *B*-factor) for the two chains (left) and the *B*-factors plotted per residue number for protein chain A (blue) and B (red) (right). (**B**) *B*-factors from the basal TLS model (left) and rms atomic fluctuations (right) in the ensemble model averaged per residue. Differences in crystal packing restrict the flexibility of chain B around residue 47. (**C**) Comparison (left) and superposition (right) of a region of the protein (indicated by black box in (**A**)) of the ensemble of structures observed for protein copy A (blue) and B (red). Analogous analyses for 2R8Q, 1YTT, 1IEP and 2XFA are shown in ***Figure 7—figure supplements 1–4***. The protein copies in 3GWH and 3ODU showed backbone shifts greater than 4.5 Å and were left out of this analysis.

The following figure supplements are available for figure 7.

**Figure supplement 1**. Comparison of atomic fluctuations for NCS related protein copies for dataset 2R8Q.

**Figure supplement 2**. Comparison of atomic fluctuations for NCS related protein copies for dataset 1YTT.

**Figure supplement 3**. Comparison of atomic fluctuations for NCS related protein copies for dataset 1IEP.

**Figure supplement 4**. Comparison of atomic fluctuations for NCS related protein copies for dataset 2XFA.

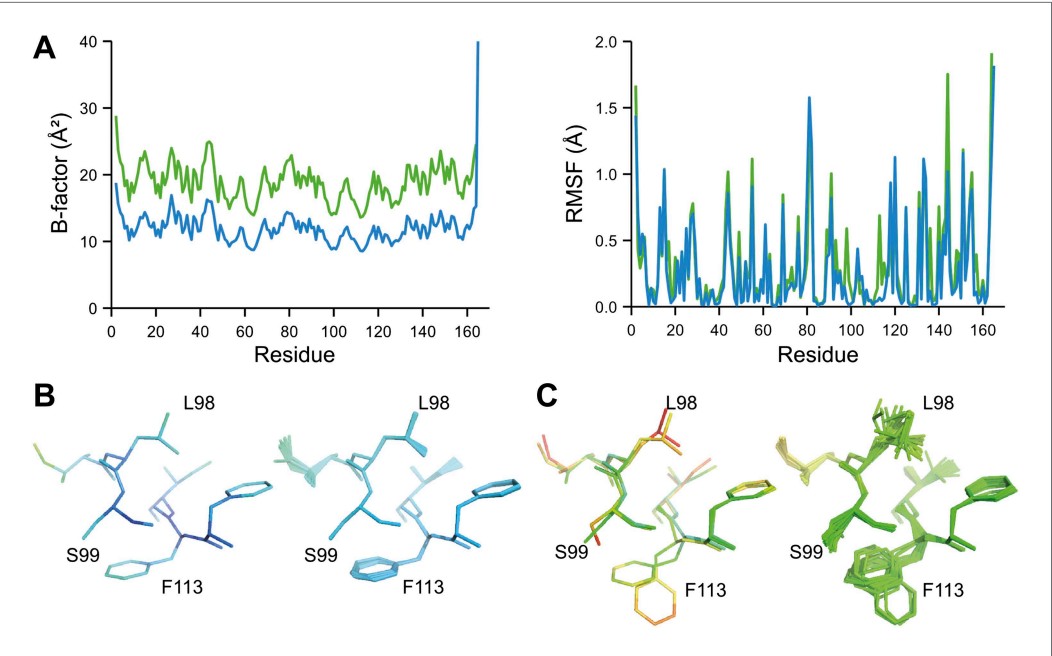

**Figure 8**. Ensemble refinement of two isomorphous proline isomerase datasets collected at 100 K and 288 K. (**A**) Left, basal TLS *B*-factors of ensemble models for 100 K and 288 K datasets (blue and green, respectively). Right, atomic rms fluctuations of ensemble models for 100 K and 288 K datasets (blue and green, respectively). (**B**) Re-refined single-structure (left) and ensemble model (right) for 100 K dataset. (**C**) Re-refined single-structure and ensemble model for 288 K dataset. In (**B**) and (**C**) atoms are coloured by *B*-factor (5 to 25 Å²). As with the published single structure refinement (***Fraser et al., 2009***) alternative conformations were not found for residues Leu98, Ser99 and Phe113 at 100K.

X-ray gradient dictates the MD sampling to yield featureless, $(mF_{obs}-DF_{model})\exp[i\varphi_{model}]$, electron-density difference maps, while the global disorder model is accounted for by taking $B_{TLS}$ into account when computing the atomic densities. In this way, the ensemble of structures is generated to model the anisotropic and anharmonic electron-density distributions precisely, while being restrained by the bonded and non-bonded energy terms used in the MD simulation. The separation of global disorder and local atomic fluctuations contrasts the original approach by ***Gros et al. (1990)***, where the MD sampling had to account for both the large scale global disorder and local fluctuations leading to very long relaxation times $\tau_x$ of 16 ps. In the current work much shorter relaxation times of 0.25–2 ps can be used, thereby limiting potential over-fitting markedly. The method is applicable to data with a wide range of upper resolution limits. We see marked improvements in $R_{free}$ for datasets ranging from 1.5 to 2.6-Å upper-resolution limit. A detailed interpretation of the ensembles is allowed, supported by the very high local correlations between independent ensemble refinements. However, at lower resolution limits and for highly disordered loops the local correlation between independent runs drops and detailed interpretation is not feasible. Thus, even though the number of independent parameters in an ensemble model is not clearly defined (and therefore the parameter-to-observation ratio is unclear), the gain in $R_{free}$ and the very high local correlations between independent runs indicate a high reliability of the ensemble models. However, the method is not a panacea for highly disordered protein regions. In the absence of ordered conformations for a certain region of the protein (as implicitly defined by the diffraction data) the ensemble refinement will sample diverse conformations in order to prevent the build up of negative peaks in the electron-density difference map. In other words, if the data 'says' that a region is disordered, ensemble refinement will generate diverse conformations for that region. Furthermore, dataset pathologies caused, for example, by radiation damage may have confounded effects that obscure the dynamics inherent to the protein molecule. Thus, perhaps somewhat counter-intuitively, this modelling method that accounts for inherent protein

**Figure 9**. Overview of side-chain dynamics in ensemble structures. Atoms are coloured by their relative probability in the ensemble (see 'Materials and methods'), reflecting the degree of disorder (ranging from well-ordered in blue to disordered in red). Bottom left insert shows secondary structure cartoon. Three datasets exhibit disordered interior sides chains forming a molten core region. (**A**) 3CA7 shows an ordered core with disordered hydrophilic side chains on the outside and is typical of the majority of the datasets. (**B**) 1BV1, the major pollen allergen and putative plant steroid transporter, has a disordered central cavity (location of cavity show with dotted lines). (**C**) 1X6P in the monomeric form of the fibril forming PAK pilin shows multiple disordered aliphatic and aromatic side chains in the interface between the N-terminal α-helix and the four stranded β-sheet domain. (**D**) Proline isomerase exhibits a molten core at 288 K, 3K0N (left); however, these interior dynamics are frozen-out at 100 K, 3K0M (right).

dynamics does not help to resolve structural details of disordered regions, but is particularly suited to resolve dynamical fluctuations in ordered parts of the protein structure.

Ensemble refinement of 20 protein datasets highlighted global dynamics features of protein molecules. Surprisingly, in some cases the ensembles indicated the existence of folded protein structures that display molten cores. Most likely, such molten cores may indicate intermediates of protein molecules that function in larger complexes (such as PAK pilin), or alternatively these molten

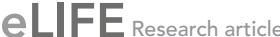

**Figure 10**. Dynamics in the binding pocket of proline isomerase at 288 K. (**A**) The location of the binding pocket comprised of residues Arg55, Met61, Ser99 and Phe113. (**B**) Zoom in of binding pocket (as dotted lines in (**A**)) showing flexible β-sheet for C=O·HN network of residues 55-62-113-98 in neighbouring β-strands. (**C**) All four residues show a ~9:1 ratio between major and minor conformations which is in good agreement with NMR relaxation dispersion data collected a similar temperature (***Eisenmesser et al., 2005***). Histograms show mean $\chi_1$ angles generated from 10 random number repeats of ensemble refinement (error bars ±1 $\sigma$). Inserts show the relevant side chains, coloured by atomic probability (see 'Materials and methods'), as observed in the ensemble reported in **Table 1**.

cores support dynamical fluctuations that are needed for ligand binding and enzyme functioning (as for birch pollen allergen and proline isomerase respectively). Furthermore, the ensembles show details of specific order–disorder transitions, or conformational exchanges, between active site and core residues (as for HIV protease in the unbound and bound state) that suggest a mechanism of entropy compensation to support the enzymatic activity. The difference in dynamics observed

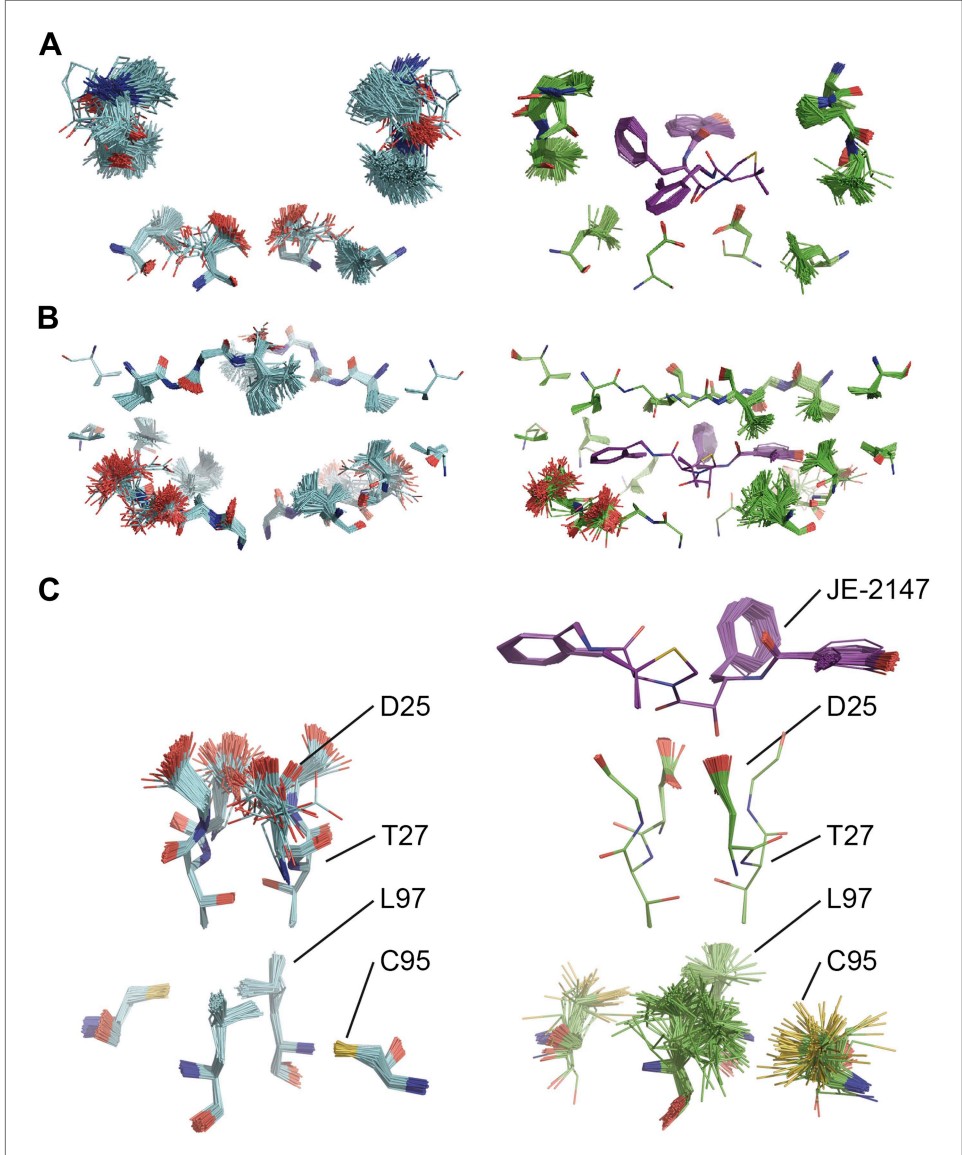

**Figure 11**. Comparison of ensemble structures of bound and unbound forms of HIV protease. (**A**) Residues in the P1 binding sites are disordered in the unbound HIV protease (2PC0), left-hand side, with carbon atoms shown in cyan, oxygen red and nitrogen blue. These residues become ordered in HIV protease in complex with a high affinity inhibitor, JE-2147 (1KZK), right-hand side with carbon atoms of the protease shown in green and of the inhibitor in purple. In 1KZK the two chains of the functional dimer are present in the asymmetric unit, whereas in 2PC0 a monomer is present in the asymmetric unit and the dimer is drawn using the crystallographic twofold axis. (**B**) Shows an alternative orientation showing the P2 binding site. (**C**) The catalytic Asp25 becomes ordered upon binding of the inhibitor, forming a hydrogen bond with the P1 carbonyl and hydroxyl of JE-2147. In contrast, the distal residues Cys95 and Leu97 at the dimer interface become less ordered upon binding.

between the ensembles of proline isomerase at cryo and ambient temperatures indicates that flash freezing of a crystal anneals local conformational fluctuations and thereby removes protein dynamics that may be functionally relevant.

In conclusion, this new method of modelling X-ray diffraction data reveals a wealth of detailed information about the dynamics of biomolecules that complements the high-resolution structural information already available from the crystallographic experiment. In depth understanding of structure–dynamics in biomolecules will enhance our insights into the molecular mechanisms that underlie biological processes.

**Figure 12**. ABL-kinase Imatinib binding site. (**A**) Imatinib binding site in chain A of the 1IEP dataset showing distribution of the six protein–ligand hydrogen bonds in chain A and chain B (red and blue respectively). (**B**) Hydrogen bond network of ordered water network observed in the re-refined single structure, left, and the ensemble model, right. (**C**) The activation loop (shown in pink) is disordered when ABL-kinase is complexed with Imatinib (shown in cyan) as observed previously in solution (***Vajpai et al., 2008***).

## Materials and methods

The method of ensemble refinement was implemented in the Phenix software (***Adams et al., 2010***). Adaptations and new procedures developed for ensemble refinement are given in section 'Ensemble

refinement methods'. Simulations were performed as described in section 'Ensemble refinement protocol'. Details of the single-structure re-refinements used for comparison with the ensemble models are given in section 'Single structure re-refinements'. Validation of the global disorder TLS model, the dependency of ensemble refinement on the starting structure and additional ensemble refinement calculations are given in 'Additional ensemble refinement calculations'.

Ensemble refinement was performed using phenix.ensemble_refinement, as will be made available in the next release of Phenix.

## Ensemble refinement methods

### Time-averaged restraints

The overall model structure factors are calculated as (1), defined by *Afonine et al. (2005)*, incorporating overall anisotropic scaling (*Sheriff and Hendrickson, 1987*) and bulk solvent contributions (*Jiang and Brünger, 1994*).

$$F_{model} = ke^{\left(-\frac{h^T A^{-1} B_{cart} \left(A^{-1}\right)^T h}{4}\right)} \left(F_{calc} + k_{sol} e^{\left(-\frac{B_{sol} s^2}{4}\right)} F_{mask}\right), \quad (1)$$

where, $k$ is the overall scale factor, $h$ is the column vector with Miller indices, $A$ is the orthogonalisation matrix, $B_{cart}$ is the anisotropic scale matrix, $F_{calc}$ is the structure factors calculated from atomic model, $k_{sol}$ and $B_{sol}$ are the parameters for the flat bulk solvent model and $F_{mask}$ are the structure factors calculated from bulk solvent mask.

In order to restrain the instantaneous structures produced during the MD simulation with time and spatially averaged X-ray data, time-averaged restraints are used (*Gros et al., 1990*). This produces time-averaged (or rolling-average) structure factors such that (1) becomes (2).

$$F_{model}^t = ke^{\left(-\frac{h^T A^{-1} B_{cart} \left(A^{-1}\right)^T h}{4}\right)} \left(\left\langle F_{calc}\right\rangle_t + k_{sol} e^{\left(-\frac{B_{sol} s^2}{4}\right)} \left\langle F_{mask}\right\rangle_t\right). \quad (2)$$

This is a time-dependent memory function, that is a 'rolling' average, where the size of the averaging window is controlled by the $\tau_x$ parameter (typically 1 ps). This averaging function is updated with the current individual structure every 10 time-steps ($\Delta t$) during the simulation and is implemented as in (3).

$$\left\langle F_{calc}\right\rangle_t = e^{-\Delta t/\tau_x} \left\langle F_{calc}\right\rangle_{t-\Delta t} + \left(1 - e^{-\Delta t/\tau_x}\right) F_{calc}^t. \quad (3)$$

### Dual explicit-bulk solvent model

Due to the stochastic behaviour of solvent molecules and the number of partially disordered or low occupancy sites, explicitly modelled solvent atoms are repositioned every 250 time-steps. Electron-density and difference density maps are generated using $F_{model}^t$, excluding reflections in the free $R$ set. Water oxygen atoms with an electron-density peak >1.0 σ in the $2mF_{obs} - DF_{model}$ map or a peak >3.0 σ in the $mF_{obs} - DF_{model}$ map are preserved, otherwise the atom is removed. New water atoms are added for positions which have a $2mF_{obs} - DF_{model}$ peak >1.0 σ and a $mF_{obs} - DF_{model}$ peak >3.0 σ, and are between 1.8–3.0 Å in distance to an existing atom. For high-resolution cases these criteria are adjusted to include $mF_{obs} - DF_{model}$ map peaks >2.5 σ. Newly positioned atoms are assigned a random, Boltzmann-weighted, velocity. Explicitly modelled solvent atoms contribute to the atomic model ($F_{calc}$).

Bulk solvent is modelled using a solvent mask (*Afonine et al., 2005*). The mask structure factors ($F_{mask}$) are averaged in the same manner as the atomic model ($F_{calc}$) (4).

$$\left\langle F_{mask}\right\rangle_t = e^{-\Delta t/\tau_x} \left\langle F_{mask}\right\rangle_{t-\Delta t} + \left(1 - e^{-\Delta t/\tau_x}\right) F_{mask}^t. \quad (4)$$

The $k_{sol}$ and $B_{sol}$ bulk solvent parameters and $B_{cart}$ scaling parameters used for the duration of the simulation are calculated from the starting structure as described previously (**Afonine et al., 2005**), they are re-optimized for the final ensemble.

## Constrained target functions

The overall scale factor, $k$, is constrained during the simulation. For the maximum-likelihood target function, as shown for acentric reflections, (5) during normalisation (6), the sum of the rolling-average structure factor array (2) is scaled to the sum of the structure factor array from the starting model ($F_{ref}$) as shown in (7).

$$P_{x\text{-}ray} = \frac{2E_{obs}}{1-\sigma_A^2}\exp\left(-\frac{E_{obs}^2 + \sigma_A^2 E_{model}^{t\,2}}{1-\sigma_A^2}\right)I_0\left(\frac{2E_{obs}\sigma_A E_{model}^t}{1-\sigma_A^2}\right), \qquad (5)$$

$$E_{model}^t = \frac{kF_{model}^t}{\sqrt{\varepsilon\,\Sigma_N}}, \qquad (6)$$

$$k = \frac{\sum\limits_{hkl} F_{ref}}{\sum\limits_{hkl} F_{model}^t}, \qquad (7)$$

where, $E_{obs}$ and $E_{model}$ are the normalised structure factors, $\sigma_A$ is the Sigma-A weighting factor, $I_0$ is a modified Bessel function of order 0 and $\varepsilon$ is the expected intensity factor.

## Temperature bath and X-ray weight

The simulations are performed such that the non-solvent atoms are at a target temperature ($T_{target}$) of 300 K, where the simulation is coupled to a velocity-scaled temperature-bath (**Berendsen et al., 1984**). The temperature bath is set to a value less than 300 K, typically 295–298 K. Because the X-ray restraints are computed from a time-dependent memory function, the X-ray energy term is non-conservative and thus heating occurs. During the equilibration phase the X-ray weight ($w_{x\text{-}ray}$) is modulated by the temperature of the protein atoms ($T_{protein}$) every 10 time-steps ($\Delta t$), such that the non-solvent atoms sample consistently at the target temperature (8).

$$w_{x\text{-}ray}^t = w_{x\text{-}ray}^{t-\Delta t}\frac{T_{target}}{T_{protein}}. \qquad (8)$$

Thus, the thermostat offset controls the X-ray weight in a system independent manner whilst maintaining the target temperature. In the acquisition phase the X-ray weight is fixed to the averaged value used in the equilibration phase.

## TLS approximation of the global disorder

The partitioning of inter-molecular disorder is performed before the start of the simulation using ADPs from the traditionally refined starting structure. TLS groups are assigned per molecule or domain as appropriate to model global packing disorder. For each group, TLS parameters are fitted to the ADPs of the starting structure for all non-solvent, non-hydrogen atoms. The agreement of the isotropic equivalents for the fitted TLS ADPs ($B_{tls}$) and the reference ADPs ($B_{ref}$) is scored as (9) for all non-solvent, non-hydrogen atoms.

$$R_i = \left| B_{ref}^i - B_{tls}^i \right|. \qquad (9)$$

A percentile of atoms with the poorest fitting ADPs ($p_{TLS}$) are excluded from the next round of TLS parameter fitting and repeated until the fitted TLS parameters converge. The converged TLS parameters are then applied to all atoms within that group for the duration of the simulation. Solvent atoms are assigned to the TLS group of the closest non-water atom, this assignment is updated every 250 time-steps. This TLS model produced lower $R$-values than using the ADP values from the re-refined single structure or using the one overall isotropic $B$-factor for all atoms in the model (**Table 5**).

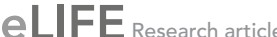

**Table 5.** Comparison of three *B*-factor models for ensemble refinement. Burling et al. (***Burling and Brunger, 1994***) had shown previously that the choice of ADPs for ensemble refinement can affect the resultant structures. Three alternative ADP models were tested for seven datasets. (1) 'Global isotropic *B*-factor', one overall isotropic *B*-factor applied to all atoms in the simulation. Multiple trials were performed to establish the optimum single value. For comparison the Wilson *B*-factor of the data is listed. (2) 'Refined ADPs', ADPs from the refined single-structures. Best results were obtained by multiplying the refined ADPs by given scale factor. (3) 'Fitted TLS ADPs', fitted TLS model obtained as described in 'Materials and methods'

| | | Global isotropic *B*-factor | | | | Refined ADPs | | | Fitted TLS ADPs | | |
|---|---|---|---|---|---|---|---|---|---|---|---|
| PDB | Resolution (Å) | $R_{work}$ | $R_{free}$ | Wilson *B*-factor (Å²) | Global *B*-factor (Å²) | $R_{work}$ | $R_{free}$ | Scale factor | $R_{work}$ | $R_{free}$ | pTLS |
| 3K0M | 1.3 | 0.117 | 0.147 | 12.0 | 12.0 | 0.125 | 0.146 | 0.9 | 0.103 | 0.130 | 0.3 |
| 3K0N | 1.4 | 0.121 | 0.153 | 19.1 | 19.1 | 0.126 | 0.153 | 0.9 | 0.114 | 0.133 | 0.1 |
| 1UOY | 1.5 | 0.103 | 0.148 | 10.4 | 9.4 | 0.107 | 0.144 | 0.9 | 0.101 | 0.136 | 0.3 |
| 3CA7 | 1.5 | 0.129 | 0.194 | 16.8 | 13.4 | 0.142 | 0.192 | 0.9 | 0.142 | 0.190 | 0.5 |
| 1X6P | 1.6 | 0.108 | 0.158 | 15.9 | 12.7 | 0.113 | 0.152 | 0.8 | 0.121 | 0.150 | 0.8 |
| 1F2F | 1.7 | 0.116 | 0.184 | 15.6 | 14.8 | 0.123 | 0.167 | 0.8 | 0.126 | 0.167 | 0.7 |
| 1BV1 | 2.0 | 0.125 | 0.192 | 22.6 | 18.1 | 0.135 | 0.191 | 0.8 | 0.145 | 0.182 | 0.6 |
| **Mean** | - | **0.117** | **0.168** | - | - | **0.125** | **0.164** | - | **0.122** | **0.155** | - |

### Generation of the final ensemble
Structure factors for the final ensemble are calculated from the population of collected structure as in (2) where $F_{calc}$ and $F_{mask}$ are defined as (10) and (11).

$$\langle F_{calc} \rangle_{final} = \frac{1}{n}\sum_{i=1}^{n} F_{calc}^i, \qquad (10)$$

$$\langle F_{mask} \rangle_{final} = \frac{1}{n}\sum_{i=1}^{n} F_{mask}^i. \qquad (11)$$

The acquisition phase is split into several time blocks, in each of which 250 structures are typically stored. The *R*-values of all possible contiguous time blocks are calculated and the periods with the lowest $R_{work}$ are selected. This selection reduces the $R_{work}$ by 0–1.0% (mean improvement in 0.3%). For the 1YTT dataset with high quality experimental phases, the block selection for lowest $R_{work}$ corresponds well with the overall map correlation coefficient computed between the experimentally phased map and the map derived from the ensemble model (***Figure 13***). Next, to reduce the redundancy in the number of structures in the final ensemble (during the simulation thousands of structures are collected), we calculate the smallest number of structures that reproduce the $R_{free}$ within 0.1%. This is performed by iteratively parsing the stored structures with increasing time spacing (see ***Figure 1D***). The overall and bulk-solvent scale factors are optimised for the final ensemble. The ensembles of structures are stored using the standard PDB format for multiple models, with *B*-factors listed as computed from the TLS model and overall *B*-factor scaling contributions.

### Calculation of atomic positional probability
All atoms comprising the ensemble are assigned a probability ($P_i$) based on the positional likelihood of *atom i* in a given model relative to the complete ensemble of models. $F_{calc}$ electron-density maps are calculated for each model in the ensemble and $\langle F_{calc} \rangle$ electron-density map is calculated for the complete ensemble as (10). $P_i$ is calculated as (12).

$$P_i = \frac{\rho_{\langle F_{calc} \rangle}^i}{\rho_{F_{calc}}^i}. \qquad (12)$$

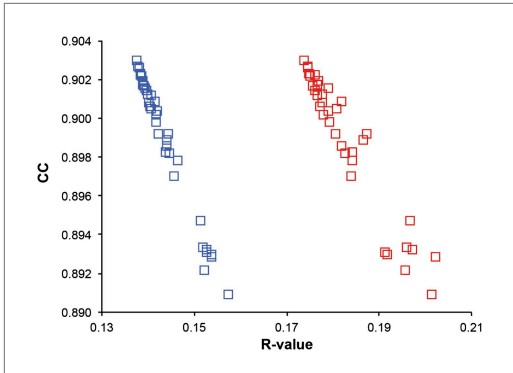

**Figure 13**. Correlation of *R*-values and overall map correlation coefficient for the 1YTT dataset in the block selection procedure. The correlation coefficients are calculated between the experimentally phased electron density map ($|\boldsymbol{F}_{obs}|\exp[i\varphi_{obs}]$) and ensemble model maps ($|\boldsymbol{F}_{model}|\exp[i\varphi_{model}]$) computed for different blocks of consecutive simulation times; blue squares indicate $R_{work}$ and red squares indicate $R_{free}$.

Calculating from an electron-density function allows for non-Gaussian distributions unlike RMSF, which is calculated from mean atomic position. These probabilities aid the visual inspection of the ensemble models and allow the observer to control the level of detail displayed (*Figure 14*).

## Ensemble refinement protocol

### Preparing the starting model

The starting structures were taken from the PDB server or from the PDB_REDO server if the $R_{free}$ was <0.25% than the equivalent PDB structure. We removed alternative positions and set corresponding occupancies to one. Overall aniso-tropic scale factors and solvent scale and *B*-factor ($k_{sol}$ and $B_{sol}$) were calculated based on these tra-ditional single-structures (i.e. using the refined *B*-factor models). Next, the atomic *B*-factors were substituted by *B*-factors derived the global TLS disorder model ('Materials and methods'— TLS approximation of the global disorder).

### X-ray restrained MD simulation

At $t$ = 0 $\langle\boldsymbol{F}_{calc}\rangle$ and $\langle\boldsymbol{F}_{sol}\rangle$ are set to $\boldsymbol{F}_{calc}$ and $\boldsymbol{F}_{sol}$. Boltzmann-weighted velocities are assigned to the atoms, corresponding to $T$ = 300 K. The bath temperature $T_{bath}$ used for velocity scaling is coupled to the X-ray weight ($w_{x\text{-ray}}$) calculation, resulting in a temperature of 300 K for all non-solvent atoms. The simulation time-step used is 0.5 fs and the force-field parameterisation is as described (*Grosse-Kunstleve et al., 2004*). Simulations are started in parallel with varying values of $p_{TLS}$ (e.g. 0.2, 0.6, 0.8, 0.9, 1.0), $\tau_x$ (e.g. 0.25, 0.5 and 1.0 ps) and $T_{bath}$ (e.g. 295 and 299 K). Water positions are picked according to electron-density criteria and updated every 250 steps. Every 10 time-steps rolling average structure factors, $\langle\boldsymbol{F}_{calc}\rangle$ and $\langle\boldsymbol{F}_{sol}\rangle$, are updated for use in the time-averaged X-ray restraints. $\sigma_A$ Values are updated if the $R_{free}$ of the rolling average model improves by >0.25%. The simulations have an equilibration phase ($20\tau_x$) in which the temperature, X-ray weight and averaged structure factors stabilize. This is followed by an acquisition phase ($40\tau_x$) where the values for $w_{x\text{-ray}}$ and $\sigma_A$ are fixed and the structures for the final ensemble model are collected.

### CPU time

CPU time for a dataset at 2.0-Å resolution with 199 residues in the asymmetric unit is 25 hr for each simulation using a 1.9 GHz processor.

## Single structure re-refinements

The single structure re-refinements used the same starting structure as the ensemble refinements (alternative conformations were not removed) and were re-refined using phenix.refine (version 1.7.1) (*Afonine et al., 2012*) and Buster (version 2.10.0) (*Bricogne et al., 2009*). Standard parameters were used with the exception of optimizing the target weights and increasing the number of macro-cycles to 8 in phenix.refine. Explicit water refinement was performed, anisotropic ADPs were used if present in starting structure and TLS parameters were defined as used in the starting structure. PDB_REDO (*Joosten et al., 2010*) models were used as deposited.

## Additional ensemble refinement calculations

### Testing *B*-factor model in ensemble refinement

Burling et al. (*Burling and Brunger, 1994*) had previously shown that the choice of ADPs for ensemble refinement can affect the resultant structures. Three alternative ADP models were tested for seven datasets, as shown in *Table 5*. ADP model 1, 'Global isotropic *B*-factor', uses one overall isotropic *B*-factor applied to all atoms in the simulation. Multiple trials were performed to establish the optimum single value. For comparison the Wilson *B*-factor of the data is listed. ADP model 2, 'Refined ADPs', uses the ADPs from the refined single-structures. Best results were obtained by multiplying the refined ADPs by given scale factor. ADP model 3, 'Basal TLS ADPs', uses the basal TLS model with one TLS

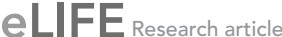

**Figure 14**. Interpretation of global and local details of 1UOY ensemble model is aided by relative atomic probability (as described in 'Materials and methods'). Ensemble models, left and centre, are colour by individual atom probability (0–1) from red to blue. Single structures, right, are coloured by individual atomic *B*-factor as refined in phenix.refine. (**A**) Global structure, selecting different probability ranges highlights partially ordered water positions. (**B**) Atomic probabilities of loop regain features correlate with *B*-factors in single structure. Anharmonic motion of Ser5 can be observed as well as anisotropic motion at Tyr7, which is shown in more detail in (**C**).

group per chain (including all non-hydrogen, non-solvent atoms) obtained as described in 'Materials and methods'—TLS approximation of the global disorder, where $p_{TLS}$ is the percentage of atoms included the iterative fitting procedure. The basal TLS model returns the lowest $R_{free}$ values in all test cases.

## Effect of starting model on ensemble refinement

To test the effect of the starting structure three datasets (1UOY, 3CA7 and 1BV1) were re-refined with Buster, phenix.refine and Refmac as given by the PDB_REDO server. Each of these re-refined structures was used as the input structure for ensemble refinement, using the same run-time parameters. Each ensemble refinement was repeated three times using a different random number to generate the

initial atomic velocities. The results are shown in *Table 3*. The mean $R_{free}$ (averaged over the random number seed repeats) of the resulting ensembles from the three different input structures are within 0.5%. The $F_{model}$ cross correlation of ensemble pairs (best representative from each program, selected by $R_{free}$) was calculated and is shown in *Table 4*. All ensemble pairs exhibit a cross correlation of greater than 0.99.

### Partial occupancies

Because occupancy and *B*-factor are strongly coupled in a traditional refinement, the occupancies of bound ligands and ions are typically set to unity, while the corresponding *B*-factors are refined in a single-structure refinement. In ensemble refinement, the *B*-factors are not refined, but are derived from the global TLS model and the atomic fluctuations.

All simulations were initially performed with full occupancy for bound ligands and ions. In several cases, this resulted in excessive sampling of the ligand or ion, as seen when inspecting the ensemble and reported by the kinetic energies during the simulation, which were far in excess of neighbouring protein atoms. These observations indicate that the corresponding occupancy of the bound ligand or ion is less than one. In these cases the occupancies were lowered and the simulations repeated until the kinetic energy of the ligand or ion were equivalent to the proximal protein components.

## Acknowledgements

We gratefully acknowledge L. Kroon-Batenburg, R.J. Read and T. Terwilliger for discussions and A.T. Brunger for providing the experimental data for the 1YTT dataset. B.T.B. and P.G. developed the method. B.T.B., with the help of P.V.A. and P.D.A., programmed the method in PHENIX. B.T.B. and P.G. analysed the data and wrote the manuscript. P.D.A. and P.V.A. assisted in the writing of the manuscript. The atomic coordinates and structure factors for the ensemble structures are available at http://www.phenix-online.org/phenix_data/ and have been deposited in the Dryad online repository (*Burnley et al., 2012*).

## Additional information

### Funding

| Funder | Grant reference number | Author |
|---|---|---|
| European Research Council | 233229 | Piet Gros |
| National Institutes of Health | P01GM063210 | Paul D Adams |
| The Netherlands Organization for Scientific Research (NWO) | 01.80.104.00 | Piet Gros |

The funders had no role in study design, data collection and interpretation, or the decision to submit the work for publication.

### Author contributions

BTB, developed the method, programmed the method in PHENIX, analysed the data and wrote the manuscript; PVA, helped program the method in PHENIX and assisted in the writing of the manuscript; PDA, helped program the method in PHENIX and assisted in the writing of the manuscript; PG, developed the method, analysed the data and wrote the manuscript

## Additional files

### Major datasets

The following datasets were generated

| Author(s) | Year | Dataset title | Dataset ID and/or URL | Database, license, and accessibility information |
|---|---|---|---|---|
| Burnley BT, Afonine PV, Adams PD, Gros P | 2012 | Data from: Modelling dynamics in protein crystal structures by ensemble refinement | http://dx.doi.org/10.5061/dryad.5n01h | Available at Dryad Digital Repository |

**The following previously published datasets were used:**

| Author(s) | Year | Dataset title | Dataset ID and/or URL | Database, license, and accessibility information |
|---|---|---|---|---|
| Reiling KK, Endres NF, Dauber DS, Craik CS, Stroud RM | 2002 | JE-2147-HIV Protease Complex | 1KZK | Publically available at the RCSB Protein Data Bank (http://www.rcsb.org/pdb/) |
| Fraser JS, Clarkson MW, Degnan SC, Erion R, Kern D, Alber T | 2009 | Cryogenic structure of CypA | 3K0M | Publically available at the RCSB Protein Data Bank (http://www.rcsb.org/pdb/) |
| Fraser JS, Clarkson MW, Degnan SC, Erion R, Kern D, Alber T | 2009 | Room temperature structure of CypA | 3K0N | Publically available at the RCSB Protein Data Bank (http://www.rcsb.org/pdb/) |
| Heaslet H, Rosenfeld R, Giffin M, Lin YC, Tam K, Torbett BE, Elder JH, McRee DE, Stout CD | 2007 | Apo Wild-type HIV Protease in the open conformation | 2PC0 | Publically available at the RCSB Protein Data Bank (http://www.rcsb.org/pdb/) |
| Olsen JG, Flensburg C, Olsen O, Bricogne G, Henriksen A | 2004 | The Bubble Protein from *Penicillium brevicompactum* Dierckx Exudate | 1UOY | Publically available at the RCSB Protein Data Bank (http://www.rcsb.org/pdb/) |
| Klein DE, Stayrook SE, Shi F, Narayan K, Lemmon MA | 2008 | High Resolution Crystal Structure of the EGF domain of spitz | 3CA7 | Publically available at the RCSB Protein Data Bank (http://www.rcsb.org/pdb/) |
| Wang H, Yan Z, Geng J, Kunz S, Seebeck T, Ke H | 2007 | Structure of LmjPDEB1 in complex with IBMX | 2R8Q | Publically available at the RCSB Protein Data Bank (http://www.rcsb.org/pdb/) |
| Bhabha G, Lee J, Ekiert DC, Gam J, Wilson IA, Dyson HJ, Benkovic SJ, Wright PE | 2011 | Crystal structure of N23PP/S148A mutant of *E. coli* dihydrofolate reductase | 3QL0 | Publically available at the RCSB Protein Data Bank (http://www.rcsb.org/pdb/) |
| Dunlop KV, Irvin RT, Hazes B | 2005 | Structure 4; room temperature crystal structure of truncated pak pilin from *Pseudomonas aeruginosa* at 1.63A resolution | 1X6P | Publically available at the RCSB Protein Data Bank (http://www.rcsb.org/pdb/) |
| Kimber MS, Nachman J, Cunningham AM, Gish GD, Pawson T, Pai EF | 2000 | SRC SH2 ThrEF1Trp Mutant | 1F2F | Publically available at the RCSB Protein Data Bank (http://www.rcsb.org/pdb/) |
| Bhabha G, Lee J, Ekiert DC, Gam J, Wilson IA, Dyson HJ, Benkovic SJ, Wright PE | 2011 | Re-refined coordinates for PDB entry 1RX2 | 3QL3 | Publically available at the RCSB Protein Data Bank (http://www.rcsb.org/pdb/) |
| Burling FT, Weis WI, Flaherty KM, Brunger AT | 1996 | Yb substituted subtilisin fragment of mannose binding protein A (Sub-MBP-A), MAD structure at 110K | 1YTT | Publically available at the RCSB Protein Data Bank (http://www.rcsb.org/pdb/) |
| Rodriguez DD, Grosse C, Himmel S, Gonzalez C, de Ilarduya IM, Becker S, Sheldrick GM, Uson I | 2009 | Crystallographic Ab Initio protein solution far below atomic resolution | 3GWH | Publically available at the RCSB Protein Data Bank (http://www.rcsb.org/pdb/) |

| Gajhede M, Osmark P, Poulsen FM, Ipsen H, Larsen JN, Joost van Neerven RJ, Schou C, Lowenstein H, Spangfort MD | 1996 | Birch pollen allergen Bet V 1 | 1BV1 | Publically available at the RCSB Protein Data Bank (http://www.rcsb.org/pdb/) |
|---|---|---|---|---|
| Nagar B, Bornmann W, Pellicena P, Schindler T, Veach DR, Miller WT, Clarkson B, Kuriyan J | 2002 | Crystal structure of the C-Abl Kinase domain in complex with STI-571 | 1IEP | Publically available at the RCSB Protein Data Bank (http://www.rcsb.org/pdb/) |
| Singh BK, Sattler JM, Chatterjee M, Huttu J, Schuler H, Kursula I | 2011 | Crystal structure of *Plasmodium berghei* actin depolymerization Factor 2 | 2XFA | Publically available at the RCSB Protein Data Bank (http://www.rcsb.org/pdb/) |
| Wu B, Chien EY, Mol CD, Fenalti G, Liu W, Katritch V, Abagyan R, Brooun A, Wells P, Bi FC, Hamel DJ, Kuhn P, Handel TM, Cherezov V, Stevens RC | 2010 | The 2.5 A structure of the CXCR4 chemokine receptor in complex with small molecule antagonist IT1t | 3ODU | Publically available at the RCSB Protein Data Bank (http://www.rcsb.org/pdb/) |
| Nagar B, Bornmann W, Pellicena P, Schindler T, Veach DR, Miller WT, Clarkson B, Kuriyan J | 2002 | Crystal Structure of the c-Abl Kinase domain in complex with PD173955 | 1M52 | Publically available at the RCSB Protein Data Bank (http://www.rcsb.org/pdb/) |
| He X, Zhou J, Bartlam M, Zhang R, Ma J, Lou Z, Li X, Li J, Joachimiak A, Zeng Z, Ge R, Rao Z, Liu Y | 2008 | A RNA polymerase subunit structure from virus | 3CM8 | Publically available at the RCSB Protein Data Bank (http://www.rcsb.org/pdb/) |
| Shimamura T, Shiroishi M, Weyand S, Tsujimoto H, Winter G, Katritch V, Abagyan R, Cherezov V, Liu W, Han GW, Kobayashi T, Stevens RC, Iwata S | 2011 | Structure of the human histamine H1 receptor in complex with doxepin | 3RZE | Publically available at the RCSB Protein Data Bank (http://www.rcsb.org/pdb/) |

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
