## [Decision Letter]

Thank you for choosing to send your work entitled “Modelling dynamics in protein crystal structures by ensemble refinement” for consideration at *eLife*. Your article has been evaluated by a Senior Editor and 3 reviewers, one of whom is a member of our Board of Reviewing Editors. The following individuals responsible for the peer review of your submission want to reveal their identity: Phil Evans.

We are pleased to report that the review of the paper has been favorable, and we invite you to revise the paper by taking into consideration the points made below.

General assessment: This paper describes a greatly improved version of time-averaging ensemble refinement that was originally developed by Gros et al., 1990. Subsequent works suggested that over-fitting may occur and it was recognized that the choice of B-factor model is very important for the success of time-average multi-conformer refinement. The current work now dramatically improves the situation by using a B-factor model that is derived from the TLS description of the single conformer isotropic B-factors. The initial implementation was requiring the dynamics simulation to model not only the local atomic fluctuations but also the effect of large-scale lattice vibrations and crystal packing defects. Taking out the large-scale effects (by modeling them with a TLS parameterization) has been crucial, and the authors show convincingly that there is no longer a problem with over-fitting. More importantly, they show that the dynamic behavior implied by the simulation correlates with other physical evidence and provides biologically relevant insights into protein function.

The new improved time-averaging method has then been tested on a set of crystal structures. Significant improvements in Rfree are reported. Specific examples illustrate the power of the method to identify sidechain motion and solvent features. Since the new improved time-averaging refinement method is now more robust than the earlier implementations, it could become a routine tool for the final refinement of macromolecular crystal structures. As such, it will then raise awareness that crystal structures are not static, but rather represent time and spatial averages. In fact, in the examples there are some surprises, of internal regions that show more flexibility than would be expected, and this is likely to have functional consequences. Thus exploring dynamics by this method may indeed produce important biological insights, at least in some cases.

Highlights:

* separation of global domain disorder from local fluctuation

* validation by comparison for a case with highly accurate MAD phases

* agreement with NMR relaxation studies for two cases

* discovery of flexible interior residues for some proteins

Points to consider in the revisions:

1. The final description of the structure as an ensemble does involve many more (non-independent) parameters than a single description, and particularly needs in essence more parameters to describe poorly ordered regions, which are the least well determined from the data. This has always been the worry about ensemble methods. The approach described here is conservative and controlled, but nevertheless is open to this criticism. It would nice to have an explicit discussion on this point.

2. In the opinion of the authors, is this a method and structure description that should be generally used? In relation to point 1, what are the dangers if the method is used by people who are less careful than the present authors?

3. It is stated that restricting “the number of structures modeled … thereby prevents over-fitting of the data”. It should be mentioned here that the 1990 work required a long relaxation time (up to 40ps) to model all the motion including that now modeled by TLS, but that the new implementation uses much smaller relaxation times (usually less than 1ps), and thus many fewer structures contribute to the running average.

4. A number of test cases are examined for the physical and biological relevance of the dynamics. It is probably not coincidental that the most interesting cases are biased towards the ones with higher resolution data. Although interesting features are seen with resolutions as low as 2.6 Angstrom, it might be worth pointing out that, because of the correlation between resolution and the number of structures that can be allowed in the ensemble average (Figure 1D), there is more potential for observing interesting features at higher resolution.

5. The discussion implies that statistics can be drawn from averaging results from several blocks each containing about 250 structures. That makes sense, since Figure 2C presents averages from more than 1,000 structures. However, a later section implies that only one acquisition period is selected for analysis. The apparent contradiction should be clarified. On this point, is there objective evidence that choosing a time block with the lowest R(work) really gives the best model? As Ian Tickle has shown, R-factors are subject to statistical error. It should not be difficult to analyze the data from the mannose-binding protein (1YTT) to see whether time blocks with low R(work) actually show the highest correlation to maps computed with experimental phases.

6. Figures 1C and 1D show a very clear trend, which the authors do not seem to comment on. The number of observations per parameter scales as 1/d(min)^3, and it seems reasonable to expect that the optimal number of structures in the ensemble average would scale in the same way because the number of parameters can be increased in line with the number of observations. Indeed, the data in Figure 1D would not fit badly to a curve described by something like 600/d(min)^3. In order for a proportional number of structures to be consulted during the ensemble averaging process, the relaxation time should also scale proportional to 1/d(min)^3, which is roughly what is seen in Figure 1C. So it looks like a very good guess of an optimal relaxation time and optimal number of structures to average could be made from just knowing the resolution.

7. Cross-validation (R_free) was used to determine the optimum relaxation time tau_x (which determines the size of the ensemble), as well as p_TLS, T_bath, and w_xray. The optimum values of tau_x are shown in Figure 1c. However, the optimum values of the other adjustable parameters should be shown as well for all cases to get a sense for the variation of these parameters for different systems.

8. Some representation (e.g., 2D contour plots) of the variation of R_free with adjustable parameters should be provided for one case (e.g., for the case with the largest improvement, 1UOY).

9. Figure 2 (1UOY): interesting case, but why is the improvement (∼ 5% in Rfree) so good compared to some of the other cases? Can this be explained?

10. For 1YTT, the improvement for the map correlation coefficient 0.895 (single conformer) to 0.903 (ensemble), and the corresponding improvement in R_free (0.014) seems small but the structures and maps (Figure 3C, D) show significant local improvements. This should be made clearer in the text, i.e., the seemingly small improvements in overall quality indicators may allow for significant local improvements. Perhaps a real space correlation plot could provide a further illustration of the improvement upon multi-conformer refinement.

11. “Similar to Brunger, the R_free and w_xray correlated with stereochemical quality… ”. This statement is unclear. R_free is not always correlated with w_xray. The weaker w_xray, the tighter geometry one would expect but with increasing R_free. Perhaps the correlation between R_free and model quality could be shown for the ensemble refinements performed?

12. Figures 7D and 8, the proline isomerase cyclophilin A, CypA. Please provide a plot with the ensemble variability (rmsd), NMR relaxation parameters (Figure 2 in Eisenmesser et al, Science, 2002), and the ADP as a function of residue. Such a plot would be an independent validation of the method by comparison to solution data, and it would also illustrate that the method goes beyond single conformer individual ADP refinement.

13. Similar for case shown in Figure 9 (2PC0), detailed comparison with NMR data (if they are readily available) and comparison to single conformer ADP refinement.

14. Equation 5 is the likelihood function for acentric reflections, not centric reflections. Also in equations 5–7, the notation implies that the calculated structure factors are the instantaneous ones, whereas they should be the rolling averages.

15. In equation 4, the bulk solvent mask excludes the explicit water molecules, i.e., the explicit water molecules are considered part of the model, correct?

---

## [Author Response]

*1. The final description of the structure as an ensemble does involve many more (non-independent) parameters than a single description, and particularly needs in essence more parameters to describe poorly ordered regions, which are the least well determined from the data. This has always been the worry about ensemble methods. The approach described here is conservative and controlled, but nevertheless is open to this criticism. It would nice to have an explicit discussion on this point*.

We discuss that the number of independent parameters in the ensemble model is ill defined. However, the Rfree improvement and the very high local correlations between independent runs indicate that ensemble refinement yields reliable models.

*2. In the opinion of the authors, is this a method and structure description that should be generally used? In relation to point 1, what are the dangers if the method is used by people who are less careful than the present authors*?

We show that the method markedly improves Rfree for a range of datasets with upper resolution limits from 1.5 to 2.6 Å resolution. It is rather difficult for us to judge what the dangers might be when used by people who are not careful. We will provide our protocols with the release of the method in the Phenix package. We find two measures very informative: the improvement in Rfree and the local correlation between independent runs. For regions with low local correlation (as for highly disordered regions), detailed structural interpretations are not valid.

*3. It is stated that restricting “the number of structures modeled … thereby prevents over-fitting of the data”. It should be mentioned here that the 1990 work required a long relaxation time (up to 40ps) to model all the motion including that now modeled by TLS, but that the new implementation uses much smaller relaxation times (usually less than 1ps), and thus many fewer structures contribute to the running average*.

We have included the differences between the original and current work.

*4. A number of test cases are examined for the physical and biological relevance of the dynamics. It is probably not coincidental that the most interesting cases are biased towards the ones with higher resolution data. Although interesting features are seen with resolutions as low as 2.6 Angstrom, it might be worth pointing out that, because of the correlation between resolution and the number of structures that can be allowed in the ensemble average (Figure 1D), there is more potential for observing interesting features at higher resolution*.

We implicitly addressed this issue by discussing the very high local correlations observed at high resolution versus the lower local correlation observed at lower resolution, and for highly disordered loops.

*5. The discussion implies that statistics can be drawn from averaging results from several blocks each containing about 250 structures. That makes sense, since Figure 2C presents averages from more than 1,000 structures. However, a later section implies that only one acquisition period is selected for analysis. The apparent contradiction should be clarified. On this point, is there objective evidence that choosing a time block with the lowest R(work) really gives the best model? As Ian Tickle has shown, R-factors are subject to statistical error. It should not be difficult to analyze the data from the mannose-binding protein (1YTT) to see whether time blocks with low R(work) actually show the highest correlation to maps computed with experimental phases*.

The apparent contradiction was removed. For the 1YTT dataset we show a strong correlation between the R values of the selected block and the overall map correlation coefficient (see Figure 11).

*6. Figures 1C and 1D show a very clear trend, which the authors do not seem to comment on. The number of observations per parameter scales as 1/d(min)^3, and it seems reasonable to expect that the optimal number of structures in the ensemble average would scale in the same way because the number of parameters can be increased in line with the number of observations. Indeed, the data in Figure 1D would not fit badly to a curve described by something like 600/d(min)^3. In order for a proportional number of structures to be consulted during the ensemble averaging process, the relaxation time should also scale proportional to 1/d(min)^3, which is roughly what is seen in Figure 1C. So it looks like a very good guess of an optimal relaxation time and optimal number of structures to average could be made from just knowing the resolution*.

We agree that there is a trend between resolution and the relaxation time (τ_x_). However, at this moment, given the limited number of datasets, we refrain from suggesting fixed limits in the parameterization of the relaxation time and currently still advise testing multiple parameters for optimum results. Future work may lead to a more deterministic parameterization.

*7. Cross-validation (R_free) was used to determine the optimum relaxation time tau_x (which determines the size of the ensemble), as well as p_TLS, T_bath, and w_xray. The optimum values of tau_x are shown in Figure 1c. However, the optimum values of the other adjustable parameters should be shown as well for all cases to get a sense for the variation of these parameters for different systems*.

As requested this information is now included in Figure 1. We note in the main text that p_TLS_ and T_bath_ do not correlate with resolution, whereas τ_x_ tends to correlate with resolution.

*8. Some representation (e.g., 2D contour plots) of the variation of R_free with adjustable parameters should be provided for one case (e.g., for the case with the largest improvement, 1UOY)*.

These have now been provided.

*9. Figure 2 (1UOY): interesting case, but why is the improvement (∼ 5% in Rfree) so good compared to some of the other cases? Can this be explained*?

1UOY exhibits a large degree of anisotropic and anharmonic side chain motion that may explain the magnitude of observed improvements. We have added a comment to the text accordingly.

*10. For 1YTT, the improvement for the map correlation coefficient 0.895 (single conformer) to 0.903 (ensemble), and the corresponding improvement in R_free (0.014) seems small but the structures and maps (Figure 3C, D) show significant local improvements. This should be made clearer in the text, i.e., the seemingly small improvements in overall quality indicators may allow for significant local improvements. Perhaps a real space correlation plot could provide a further illustration of the improvement upon multi-conformer refinement*.

We have amended the text as suggested. Figure 3 shows the real-space correlation plot of the ensemble model versus the experimentally phased map.

*11. “Similar to Brunger, the R_free and w_xray correlated with stereochemical quality… ”. This statement is unclear. R_free is not always correlated with w_xray. The weaker w_xray, the tighter geometry one would expect but with increasing R_free. Perhaps the correlation between R_free and model quality could be shown for the ensemble refinements performed*?

A figure has been added pointing out the relationship between best R_free versus w_xray and stereochemistry.

*12. Figures 7D and 8, the proline isomerase cyclophilin A, CypA. Please provide a plot with the ensemble variability (rmsd), NMR relaxation parameters (Figure 2 in Eisenmesser et al, Science, 2002), and the ADP as a function of residue. Such a plot would be an independent validation of the method by comparison to solution data, and it would also illustrate that the method goes beyond single conformer individual ADP refinement*.

*13. Similar for case shown in Figure 9 (2PC0), detailed comparison with NMR data (if they are readily available) and comparison to single conformer ADP refinement*.

Re 12&13, in the 2002 manuscript by Eisenmesser et al. the NMR relaxation data is not tabulated. Unfortunately neither proteins are listed in the BMRB database (http://www.bmrb.wisc.edu/search/query_grid/kinetic_grid.html).

*14. Equation 5 is the likelihood function for acentric reflections, not centric reflections. Also in equations 5–7, the notation implies that the calculated structure factors are the instantaneous ones, whereas they should be the rolling averages*.

The text has been corrected; the equation shown exemplifies the general case of acentric reflections. We define **F**_model_ as a function of rolling averages <**F**_calc_> and <**F**_mask_>; see equation 2. This choice simplifies the nomenclature when describing map coefficients.

*15. In equation 4, the bulk solvent mask excludes the explicit water molecules, i.e., the explicit water molecules are considered part of the model, correct*?

This is correct. We now state this explicitly.